# LESS IS MORE: HIGH-VALUE DATA SELECTION FOR VISUAL INSTRUCTION TUNING

## ABSTRACT

Visual instruction tuning is the key to building large vision language models (LVLMs), which can greatly improve the task generalization and solving capabilities by learning a mixture of instruction data from diverse visual tasks. Previous work mostly collects multiple existing visual instruction datasets via heuristic ways for training (even more than a million instructions), which may introduce data redundancy and enlarge the training cost. To investigate this issue, we conduct a series of empirical studies, which reveal a significant redundancy within the visual instruction datasets, and show that greatly reducing the amount of instructions from several tasks even do not affect the performance. Based on the findings, we propose a high-value data selection approach **TIVE**, to eliminate redundancy within the visual instruction data and reduce the training cost. In TIVE, we first estimate the instance influence score on its corresponding task, and the task difficulty score, based on the gradient-based influence functions. Then, we leverage the two kinds of scores to determine the task proportion within the selected visual instruction subset, and select high-value instances for each task, respectively. Experiments on various LVLMs show that our approach using only about 15% data can achieve comparable average performance to the full-data fine-tuned model across eight benchmarks, even surpassing it on four of the benchmarks. Our code and data will be publicly released.

## 1 INTRODUCTION

The advent of large language models (LLMs) (Brown et al., 2020; Ouyang et al., 2022; Touvron et al., 2023; Zhao et al., 2023b) has marked significant advancements in the field of artificial intelligence (AI), exhibiting excellent capabilities in human instruction following, world knowledge utilization, and complex reasoning. A surge of recent studies (Zhu et al., 2023; Liu et al., 2023b; Dai et al., 2023; Liu et al., 2023a) equip LLMs with the vision encoder to empower the capability of processing visual information. Through vision-language alignment pre-training and visual instruction tuning, *Large Vision Language Models (LVLMs)* are created to extend the application of LLMs into multimodal tasks and scenarios.

Visual instruction tuning (Liu et al., 2023b; Dai et al., 2023) is the key technique for improving the task generalization and instruction following capabilities of LVLMs, which relies on a set of visual instructions for fine-tuning. Therefore, the construction of visual instruction datasets is very crucial for LVLMs. Typically, there are two widely used ways to construct visual instructions: synthesizing instructions based on LLMs (Liu et al., 2023b) or transforming existing vision-language datasets into visual instructions (Dai et al., 2023; Liu et al., 2023a). To achieve better performance, existing LVLMs generally combine a mixture of visual instructions from different domains or tasks, to compose a large-scale visual instruction dataset. The LVLMs fine-tuned on these mixtures of visual instructions have shown remarkable performance on massive downstream multimodal benchmarks.

However, such a mixture of instructions may also introduce significant data redundancy, leading to increased training costs and potentially overfitting risk. To investigate the redundancy issue, we first conduct an empirical study on the visual instruction dataset of state-of-the-art open-source LVLM, *i.e.*, LLaVA-1.5 (Liu et al., 2023a), by reducing the instruction amount of a certain task and then evaluating the performance. The results show that the reduction of instruction data only leads to slight or even no performance decline across most benchmarks, indicating that there exists redun-

dancy within the used visual instructions. Therefore, it is promising to mitigate this redundancy by selecting a small set of representative data samples. Furthermore, we also find that the degree of redundancy varies across different tasks. It suggests that the contribution of each task should be considered when performing the redundancy elimination.

To this end, in this paper, we propose a data selection approach for visual instruction tuning, namely **TIVE**, based on *Task and Instance Value Estimation*. The key motivation is to estimate the value of each instance and then select the high-value ones, based on its influence on LVLM fine-tuning process. According to the influence function theory (Pruthi et al., 2020), the influence of an instance on the training process can be estimated by its gradient similarities with other instances. However, due to the large-scale parameters of LVLMs, the computation of gradient similarity may cause unaffordable cost. Besides, since the goal of visual instruction tuning is to learn the solving capability for diverse tasks, it is necessary to measure the influence of task learning (Pruthi et al., 2020; Xia et al., 2024), instead of only cross-instance influence.

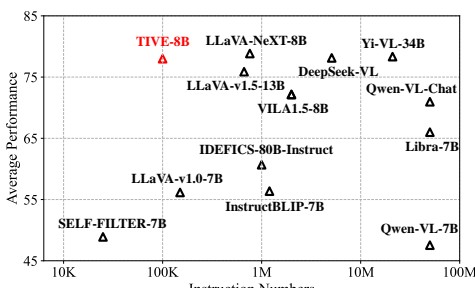

Figure 1: A comparison of TIVE-8B with other open-source models in terms of the instruction data scale and average benchmark performance on MME, SEED-Bench, MMBench, ScienceQA.

In TIVE, we adjust the gradient computation and influence estimation strategies, to better adapt into visual instruction tuning of LVLMs. To reduce the cost, we only leverage the gradients of the LoRA (Hu et al., 2021) matrices from LLM for influence estimation. These parameters are the key components for learning visual understanding and instruction following capabilities, hence their gradients would be informative features. To focus on task learning, we estimate the contribution of each instance to its corresponding task, by computing the average influence of each instance on all other in-task instances, namely *instance influence score* to help distinguish the most useful instances. Then, we measure the difficulty of each task for LVLM to learn, by computing the average self-influence of all its contained data instances, as the *task difficulty score* to help determine the task data proportion. Guided by the above scores, we can select the high-value instances to fine-tune the LVLM, for efficiently and effectively learning all the involved tasks within the visual instruction dataset.

To demonstrate the effectiveness of our approach, we apply our data selection method into several SOTA LVLMs and widely-used instruction datasets, and perform evaluation on eight benchmarks. By only using the selected 15% subset from the visual instruction dataset, the fine-tuned LVLMs can achieve comparable performance to the full-data fine-tuned model, even outperforms it on four benchmarks. As shown in Figure 1, our TIVE-8B (based on LLaVA-LLaMA3-8B) can reach the SOTA performance with much fewer instructions than SOTA methods.

## 2 REDUNDANCY ANALYSIS ON VISUAL INSTRUCTION DATA

In this section, we conduct an empirical study to examine: (1) whether data redundancy exists in existing visual instruction datasets, and (2) whether the degree of redundancy differs in different task instructions.

### 2.1 ANALYSIS SETUP

Given a mixture of visual instruction datasets for training LVLMs, we prune the amount of visual instructions from a certain task and then examine the performance change after fine-tuning with the adjusted instruction dataset. In this experiment, we mainly study the used instruction dataset for training the SOTA open-source LVLM, LLaVA-1.5 (Liu et al., 2023a).

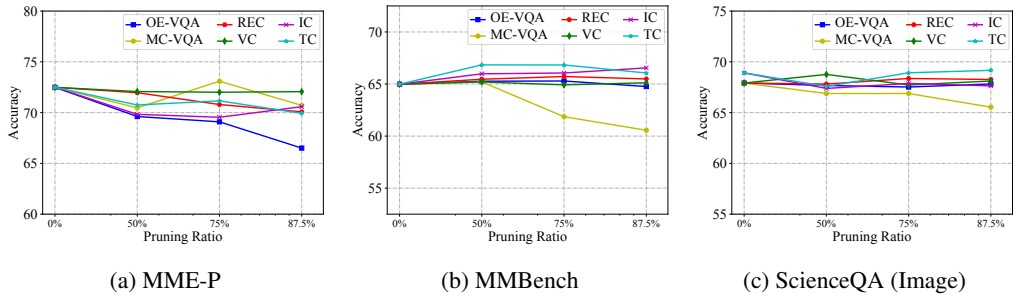

(a) MME-P        (b) MMBench        (c) ScienceQA (Image)

Figure 2: Evaluation results after pruning the amount of visual instructions from one task. Pruning 87.5% data for most tasks only leads to slight performance changes on three benchmarks.

**Backbone Model.** We choose the LLaVA-1.5 (Liu et al., 2023a) model after cross-modal alignment training as the backbone model (without instruction-tuning), which has been trained on more than 500k image-text pairs. It incorporates CLIP (Radford et al., 2021) as the visual encoder and Vicuna-v1.5 (Chiang et al., 2023) as the LLM, and further leverages two linear layers for mapping the encoded visual features to the latent space of LLM.

**Visual Instruction Dataset.** LLaVA-1.5 has been fine-tuned on a mixture of instruction datasets from different tasks. To ensure internal consistency across different tasks, only one dataset for each type of task will be selected. Appendix B contains details about the datasets. To investigate the redundancy issue in visual instruction datasets, we gradually halve the number of instructions from each task, then fine-tune the backbone model on the new instruction set and compare the performance change. For all experiments, we follow the experimental configuration of LLaVA-1.5.

**Evaluation Benchmark.** To conduct a comprehensive analysis, we evaluate the fine-tuned LVLMs on the three commonly-used benchmarks: MME-P (Fu et al., 2023), ScienceQA (Lu et al., 2022), and MMBench (Liu et al., 2023d). Details of these benchmarks can be found in Appendix C.

## 2.2 RESULTS AND FINDINGS

According to the results in Figure 2, we list the main findings as follows:

First, *there exists a significant redundancy in visual instruction datasets.* We can observe that decreasing the amount of instruction data only leads to slight performance drop in most cases. For example, reducing the number of VC would not significantly affect the model's performance across all benchmarks, and even lead to improvement on ScienceQA using 50% of data. It indicates that not all the used instruction datasets are indispensable.

Second, *for each task, the redundancy degree of different instruction datasets differs.* For OE-VQA and MC-VQA, reducing their instruction number leads to relatively significant performance degradation, *e.g.* 8% on MME-P and 7% on MMBench using a pruning ratio of 87.5%, respectively. While pruning task instructions from VC leads to minimal decline on most of the benchmarks. It indicates that different task instructions contribute to the model's final performance differently. Therefore, it is necessary to estimate the value of each task, for helping set a more proper pruning ratio and mixing proportion for all the tasks.

## 3 APPROACH

In this section, we present our approach **TIVE**, to reduce the redundancy of visual instruction data. Based on the findings in section 2, it is necessary to consider the contribution degree to the learning of diverse tasks during fine-tuning LVLMs. Specially, we consider measuring both task difficulty and instance influence scores for helping select visual instruction data. Based on the two kinds of estimated scores, we design the data selection process, to sample a small high-value visual instruction subset for efficiently and effectively fine-tuning LVLMs. We show the details of TIVE in Figure 3.

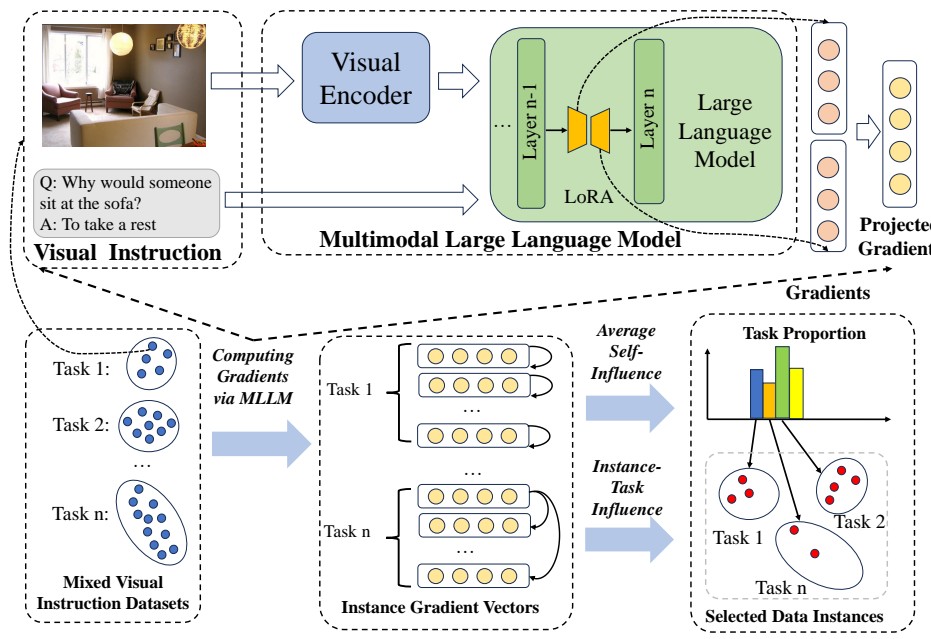

Figure 3: The illustration of our proposed approach. We utilize the gradient vectors from the LoRA parameters of the LLM, to compute the task difficulty and instance influence scores. Then, these scores are leveraged to determine the task data proportion and instance selection probability.

## 3.1 PROBLEM FORMULATION

The elimination of dataset redundancy aims to select a high-quality subset from a large dataset suffering the redundancy issue. The selected subset should contain relatively few but informative samples, to ensure the performance of the models trained on it. In this work, we focus on reducing the redundancy of the visual instruction data pool $\mathcal{D} = \{D_1, ..., D_n\}$, which is a mixture of multiple highly diverse instruction datasets from different tasks. Each dataset comprises a set of instruction samples, denoted as $D_i = \{s_1, ..., s_n\}$. Our goal is to select a data subset $\mathcal{D}_T$ from the visual instruction data pool for fine-tuning LVLMs. We use $|\mathcal{D}_T|$ to denote the target size of the selected subset.

Specially, we select the data subset from two perspectives, with the help of a pre-learned reference model trained on the sampled small set of the visual instruction data. First, we estimate the value of each task and rely on its difficulty to determine their proportions within the final subset $\mathcal{D}_T$. Second, we estimate the value of each instance within each task $D_i$ to select the most useful instances for this task.

## 3.2 ESTIMATING TASK DIFFICULTY AND INSTANCE INFLUENCE

In this part, we present how we measure the task difficulty and instance influence scores based on the influence on fine-tuning LVLMs. According to the influence formulation (Pruthi et al., 2020), the influence of a training instance $s$ on the another instance $s'$ can be denoted as:

$$\text{Inf}(s, s') = \nabla l(s, \theta) \cdot \nabla l(s', \theta), \tag{1}$$

where $\theta$ and $\nabla l(s, \theta)$ denote the parameters of the LVLM and their gradients, respectively. Based on it, we devise two formulations for estimating the influence of each instance on learning its corresponding task, and measuring the difficulty of learning each task, respectively.

**Instance Influence Estimation.** To efficiently learn each task during visual instruction tuning, we aim to obtain the contribution of each task instance, to help select a small proportion of training

samples which are highly important for the task learning. Our motivation is that if an instance has a higher positive influence on the learning of all other instances within the task, it can be regarded as a higher-value instance for helping learn the task and should be selected. Therefore, given an instance $s$ from task set $D_i$. we compute the average influence of the instance on all other instances from its affiliated task, denoted as:

$$v_s^i = \frac{1}{|D_i|} \sum_{s' \in D_i \setminus s} \frac{\nabla l(s, \theta) \cdot \nabla l(s', \theta)}{|\nabla l(s, \theta)||\nabla l(s', \theta)|}. \tag{2}$$

We normalize the gradients to mitigate the impact caused by abnormally large gradient values. By this way, we can compare the influence of different instances within each task, and select the high-value ones for training. However, such algorithm relies on a double-loop to compute the instance influence for all samples within the task data $D_i$. An alternative is to first compute the average gradient across all samples (which can be regarded as the task gradient), then compute the instance influence for each sample, denoted as:

$$v_s^i = \frac{\nabla l(s, \theta)}{|D_i||\nabla l(s, \theta)|} \cdot \sum_{s' \in D_i} \frac{\nabla l(s', \theta)}{|\nabla l(s', \theta)|}. \tag{3}$$

In this way, we can compute the instance influences for all samples within the task data through two forward passes, thereby enhancing computational efficiency.

**Task Difficulty Estimation** According to our findings in section 2, the impact of pruning different task instruction amount also differs in the LVLM performance. It is because not all the involved tasks are so hard that require such number of training instances, and it is promising to prune their data amount for reducing redundancy. Therefore, we aim to measure the difficulty of all the tasks within the visual instruction dataset, to adjust their proportion in the selected subset. Concretely, we employ the average self-influence score of all the in-task instance, to measure the task difficulty. Self-influence is to estimate the influence of training an instance on learning itself, denoted as $\nabla l(s, \theta) \cdot \nabla l(s, \theta)$. A higher self-influence score indicates that the instance is hard to learn (Bejan et al., 2023), as it leads to large gradient values. By averaging the self-influence scores of all instances from each task, we can estimate the overall difficulty of a task as:

$$v_i^t = \frac{1}{|D_i|} \sum_{s \in D_i} \nabla l(s, \theta) \cdot \nabla l(s, \theta). \tag{4}$$

Based on the task difficulty score, we can determine the proportion of all the task data within the selected visual instruction subset. In this way, the difficult task should be assigned with a larger proportion of selected data, while the redundant data within the easy tasks should be removed more.

**An Efficient Estimation for Both Data Values** In addition to the above estimation methods, we develop a more efficient implementation for TIVE, TIVE-efficient, to reduce the time cost for gradient computation across all samples. To achieve this, we uniformly sample a small amount of instances for each task and compute their gradients. Based on these gradients, we can estimate task difficulty and instance influences using the above algorithm. Then, using the instance influences of these samples, we train a small model to predict the instance influences for remaining samples. In this way, we significantly reduce the time cost for gradient computation, while slightly compromising estimation accuracy. TIVE-efficient can serve as an effective alternative when computational resources are limited.

### 3.3 DATA SUBSET SELECTION

In this section, we introduce how we obtain the gradient features and select a small data subset based on the proposed data value measurements.

**Gradient Features Computation.** Firstly, to efficiently compute the gradient features, we train a reference model with LoRA (Hu et al., 2021) using a small amount of instruction data. In this

Table 1: A comparison between TIVE and other baseline approaches for data selection on several downstream benchmarks. Benchmark names are abbreviated due to space limits. MME-P: MME-Perception, MME-C: MME-Cognition, SEED-I: SEED-Bench (Image), MMB: MMBench, MMB-CN: MMBench (Chinese), SQA: ScienceQA, SQA-I: ScienceQA (Image). * indicates our reimplemented results. Rel. represents the average relative performance compared to baseline model. Improvement over best represents the relative improvement of TIVE over the best performance among other baseline approaches. **Bold** and underline fonts indicate the best and second best performance on the task.

| Method | # Ins | MME-P | MME-C | SEED-I | MMB | MMB-CN | SQA | SQA-I | POPE | Rel. |
|---|---|---|---|---|---|---|---|---|---|---|
| BLIP-2 | - | 1293.8 | - | - | - | - | - | 61.0 | 85.3 | - |
| InstructBLIP-7B | 1.2M | - | - | - | 36.0 | 23.7 | - | 60.5 | - | - |
| Shikra | 5.5M | - | - | - | 58.8 | - | - | - | - | - |
| IDEFICS-80B | 1M | - | - | - | 54.5 | 38.1 | - | - | - | - |
| Qwen-VL | 50M | - | - | - | 38.2 | 7.4 | - | 67.1 | - | - |
| Qwen-VL-Chat | 50M | 1487.5 | - | - | 60.6 | 56.7 | - | 68.2 | - | - |
| InstructionGPT-4 | 0.2K | 463.3 | - | - | 31.4 | - | - | - | - | - |
| SELF-FILTER | 25K | 955.6 | - | 47.5 | 38.5 | - | 59.4 | - | - | - |
| **Backbone model** | | | | | | | | | | |
| LLaVA-1.5 | 665K | **1510.7** | 311.9* | **66.1** | 64.3 | **58.3** | 69.4* | 66.8 | **85.9** | 100.0% |
| **Our experiment** | | | | | | | | | | |
| Random | 100K | 1386.5 | 271.3 | 61.9 | 61.8 | 54.5 | 69.8 | 68.4 | 83.9 | 95.2% |
| Length | 100K | 1413.0 | 266.1 | 61.2 | 59.3 | 53.9 | 71.1 | 69.2 | 83.3 | 94.8% |
| Perplexity | 100K | 1393.3 | 260.7 | 61.3 | 62.3 | 55.0 | 70.5 | 67.9 | 83.6 | 94.9% |
| GraNd | 100K | 1400.5 | 287.1 | 62.3 | 62.9 | 54.3 | 71.4 | 68.4 | 82.5 | 96.3% |
| EL2N | 100K | 1356.5 | 294.7 | 61.9 | 61.6 | 56.1 | 70.2 | 66.2 | 84.6 | 95.5% |
| MoSo | 100K | 1410.2 | 288.5 | 62.4 | 62.6 | 55.4 | 69.8 | 68.1 | 83.6 | 96.4% |
| LESS | 100K | 1415.1 | 279.1 | 62.2 | 63.0 | 55.8 | 71.2 | 68.8 | 84.1 | 96.7% |
| TIVE (ours) | 100K | 1433.0 | **322.1** | 63.2 | **65.0** | 58.2 | **72.2** | 70.6 | 85.6 | 100.3% |
| TIVE-efficient (ours) | 100K | 1424.9 | 315.3 | 62.5 | 64.3 | 58.2 | 72.1 | **70.8** | 85.4 | 99.9% |

way, the reference model can be warm-up to learn the visual instruction following capability, and has not overfitted to the distribution of the whole visual instruction dataset. Thus, the gradients from the reference model can store useful information about visual instruction tuning for following influence estimation. After training the reference model, we can obtain the gradient features through backward propagation. To save storage and computation, we follow existing work (Pruthi et al., 2020) to reduce feature dimensions with random projection. Such projection often preserves the inner products (Johnson, 1984), ensuring the effectiveness of the projected gradient features.

**Selecting Data based on Estimated Values.** After obtaining the task-level and instance-level data values, we can select the subset from the visual instruction data pool. First, we use the task-level value to determine the proportion for each task in the data subset. The target data subset $\mathcal{D}_T = \{D_1^{'}, ..., D_n^{'}\}$ contains the same number of task datasets as the original data pool, but changes the total amount and task proportion. For each task subset $D_i^{'}$, we compute its data proportion within the target data subset as $p_i^{'} = \frac{v_i^t}{\sum_{j=1}^{n} v_j^t}$. where $v_i^t$ is the estimated task-level value. Then, we rely on the instance-level value to sample $|D_i^{'}|$ instances from the original visual instruction dataset. Here, we directly employ the softmax function $w_i = \text{softmax}(v^i/\lambda)$ to map the instance-level value to a sampling weight distribution. We use a hyperparameter $\lambda$ to control the temperature of the weight distribution. For all the tasks, we sample the instances based on the above weight distribution, and merge all the datasets to compose our final selected data subset.

# 4 EXPERIMENTS

## 4.1 EXPERIMENT SETUP

We conduct extensive experiments on TIVE across various models and datasets. More information about the training datasets, baselines, evaluation benchmarks, and implementation details are presented in Appendix B, Appendix D, Appendix C, and Appendix E, respectively.

Table 2: The performance of TIVE across different LVLMs. # Samp indicates the sampling ratio.

| Model | Method | # Samp | MME-P | MME-C | SEED-I | MMB | SQA | SQA-I | POPE | Rel. |
|-------|--------|--------|-------|-------|--------|-----|-----|-------|------|------|
| LLaVA-Vicuna-7B | - | 100% | **1510.7** | 311.9 | **66.1** | 64.3 | 69.4 | 66.8 | **85.9** | 100.0% |
| | Random | 15% | 1386.5 | 271.3 | 61.9 | 61.8 | 69.8 | 68.4 | 83.9 | 95.2% |
| | TIVE | 15% | 1433.0 | **322.1** | 63.2 | 65.0 | **72.0** | 70.6 | 85.6 | 100.3% |
| | TIVE | 30% | 1467.2 | 309.8 | 64.4 | **66.5** | 71.4 | 70.1 | 85.2 | 100.6% |
| LLaVA-Vicuna-13B | - | 100% | 1531.3 | 295.4 | **68.2** | 67.7 | 74.4 | 71.6 | 85.9 | 100.0% |
| | Random | 15% | 1456.6 | 307.1 | 63.4 | 64.9 | 73.5 | 69.4 | 85.5 | 96.6% |
| | TIVE | 15% | 1502.9 | **336.1** | 65.3 | 66.1 | **74.5** | **72.2** | 86.3 | 100.5% |
| | TIVE | 30% | **1545.4** | 298.6 | 65.6 | **68.8** | 74.2 | **72.2** | **86.5** | 100.1% |
| LLaVA-Phi-3-4B | - | 100% | **1440.8** | 301.6 | **66.7** | 67.9 | 81.0 | 73.6 | **85.1** | 100.0% |
| | Random | 15% | 1329.1 | 295.4 | 63.1 | 64.0 | 80.2 | 71.2 | 82.8 | 95.7% |
| | TIVE | 15% | 1386.9 | 306.4 | 63.9 | 66.0 | 81.2 | 73.5 | 84.1 | 98.0% |
| | TIVE | 30% | 1425.0 | **338.2** | 65.1 | **68.5** | **81.8** | **74.3** | 83.8 | 100.9% |
| LLaVA-LLaMA3-8B | - | 100% | **1569.4** | **338.6** | 68.8 | 71.2 | 77.2 | 73.5 | **85.7** | 100.0% |
| | Random | 15% | 1495.8 | 318.2 | 65.2 | 67.9 | 80.4 | 75.4 | 83.3 | 97.4% |
| | TIVE | 15% | 1511.4 | 331.1 | 67.4 | 69.8 | **81.6** | **75.7** | 84.9 | 99.5% |
| | TIVE | 30% | 1560.3 | 322.9 | 68.1 | **72.0** | 80.5 | 74.1 | 84.6 | 100.2% |

## 4.2 Main Results

We present the comparison of TIVE with other baseline methods on LLaVA-1.5 in Table 1, the results of TIVE across different LVLMs in Table 2, and the results of TIVE across different instruction datasets in Table 3. The overall time cost of TIVE and TIVE-efficient is demonstrated in Appendix A. We present analyses of the results as follows:

**Comparison of TIVE with other baseline methods.** In Table 1, we compare TIVE with several baseline methods on 8 benchmarks. First, we observe that the traditional data selection approaches (GraNd and EL2N) perform slightly better than random selection. A possible reason is that these approaches indeed select valuable data, but are also more vulnerable to the data noise, resulting in a limited improvement. For the data selection approaches used in LLM instruction tuning (Length and Perplexity), the performances across several benchmarks are even worse than random selection. We discover that these approaches mostly focus on selecting samples which have a high influence on improving the model's generation ability, which leads to minor enhancement on the model's ability on visual understanding. It is clear that our approach significantly outperforms all other baselines and achieves consistently promising results across all benchmarks under a limited data setting. With only *15% of the instruction data*, our approach can achieve *100.3%* average performance on all benchmarks compared to the LLaVA-1.5 model, even surpass the performance of LLaVA-1.5 in four benchmarks. Simultaneously, TIVE-efficient can also surpass all baseline methods and achieve 99.9% of the performance compared to LLaVA-1.5, with negligible difference from TIVE. These results show that our proposed approach can effectively address the issues of data redundancy within LLaVA-1.5 instructions.

**Performance of TIVE across different LVLMs.** Table 2 shows the performance of TIVE on different LVLMs. We find that under the same sampling ratio (15%), our approach significantly outperforms the random baseline across all LVLMs on all benchmarks, achieving an average improvement of at least 2.3%. Simultaneously, when the sampling ratio is increased to 30%, our approach achieves better average performance than full data performance across all models, proving that TIVE successfully eliminates redundancy in visual instruction data and is effective across different LVLMs. Furthermore, we discover that under a low sampling ratio (15%), LLaVA-Vicuna-13B achieves the best average relative performance (100.5%), while LLaVA-Phi-3-4B achieves the worst (98.0%). This indicates that LVLMs with a larger LLM backbone have a relatively better average performance under less data, which is consistent with the results for LLM on language instruction tuning scenarios.

**Performance of TIVE across different instruction datasets.** We present the results of TIVE on two other instruction datasets in Table 3. We observe that TIVE remains effective on different instruction datasets. On the SVIT-Mix dataset, it significantly outperforms other baselines in five out of six benchmarks, and surpasses the full data performance in three out of the six benchmarks. On the Mini-Gemini dataset, TIVE shows more advantage over the other baseline methods, and

Table 3: The performance of TIVE across different instruction datasets.

| Method | # Samp | MME-P | MME-C | MMB | SQA | SQA-I | POPE | Rel. |
|---|---|---|---|---|---|---|---|---|
| **LLaVA-1.5** | | | | | | | | |
| Baseline | 100% | **1510.7** | 311.9 | 64.3 | 69.4 | 66.8 | **85.9** | 100.0% |
| TIVE | 15% | 1433.0 | **322.1** | **65.0** | **72.0** | **70.6** | 85.6 | 100.3% |
| Random | 15% | 1386.5 | 271.3 | 61.8 | 69.8 | 68.4 | 83.9 | 95.2% |
| Length | 15% | 1413.0 | 266.1 | 59.3 | 71.1 | 69.2 | 83.3 | 94.8% |
| **SVIT-Mix** | | | | | | | | |
| Baseline | 100% | **1443.5** | 306.1 | **67.3** | 70.2 | 68.0 | **85.3** | 100.0% |
| TIVE | 15% | 1391.7 | **306.8** | 65.8 | **72.3** | **71.2** | 84.3 | 99.8% |
| Random | 15% | 1402.9 | 288.8 | 60.2 | 69.6 | 65.7 | 83.8 | 96.0% |
| Length | 15% | 1366.5 | 301.1 | 61.3 | 70.2 | 67.1 | 84.2 | 96.8% |
| **Mini-Gemini** | | | | | | | | |
| Baseline | 100% | **1538.4** | 324.9 | **68.1** | 72.0 | 69.9 | 85.1 | 100.0% |
| TIVE | 15% | 1506.9 | **345.4** | 67.9 | **72.6** | **71.1** | **85.4** | 101.2% |
| Random | 15% | 1404.8 | 305.4 | 62.2 | 71.1 | 69.2 | 84.9 | 95.7% |
| Length | 15% | 1403.3 | 313.2 | 62.1 | 70.3 | 67.9 | 83.7 | 95.4% |

Table 4: The ablation of effectiveness of different data values. ETG indicates equal task grouping.

| Benchmarks | Ours (+Both) | +Task-level | +Instance-level w/ ETG | +Instance-level w/o ETG | Neither |
|---|---|---|---|---|---|
| SQA-I | **70.6** | 69.8 | 68.2 | 67.5 | 68.4 |
| MMB | **65.0** | 63.7 | 62.9 | 62.6 | 62.5 |
| SEED-I | **63.2** | 62.7 | 62.9 | 62.3 | 62.2 |

the average performance of TIVE on these benchmarks is better than the full data performance. Considering that the Mini-Gemini dataset has a larger number of instructions, TIVE may be more effective at eliminating redundancy when dealing with a substantial amount of instructions. These results demonstrate the effectiveness of TIVE across different instruction datasets.

## 4.3 MORE DETAILED ANALYSIS

**Effectiveness of Data Value Measurements.** We conduct a series of ablation studies to validate the efficacy of our proposed data value on both instance-level and task-level. Initially, to verify the effectiveness of task value estimation, we conduct equal task grouping by setting the proportion of all tasks in the target data subset to a equal fixed value and then conduct data selection based on instance influence only. We also add experiments where we don't consider task proportions at all and select data solely based on instance value. Subsequently, to verify the effectiveness of instance value estimation, we calculate task weights based on task difficulty, but select instances within task instructions randomly. We present our results in Table 4.

We discover that data selection based on task difficulty only or instance influence only can both boost the performance on all benchmarks. Addtionally, data selection based on task difficulty only contributes more substantially to the reduction of redundancy compared to selection based on instance influence only. Overall, selecting data based on both instance influence and task difficulty achieve the best results than all other baseline methods on all of the benchmarks, which proves the effectiveness of both values.

**Model Performance with Different Sampling ratio.** To explore the trend of model performance as data size changes, we conduct a series of experiments with different data sampling ratio. In all experiments, we maintain consistency in the data selection approach as well as model training configuration. Our experimental results are presented in Figure 4a.

As we can observe, the model's performance continuously improves with the increasing amount of data yet, the trend of this enhancement varies across different tasks. The model's performance on MME-P rapidly increases as the data size increases. However, on MMBench and SQA-I, the

Table 5: The ablation of different warm-up data size. # Avg indicates the average performance on the benchmarks. We normalize the scores on MME-P and MME-C for computing average performance.

| # Samp | MME-P | MME-C | SEED-I | MMB | SQA | SQA-I | POPE | # Avg |
|--------|-------|-------|--------|------|------|-------|------|-------|
| 2% | 1424.9 | 284.6 | 63.1 | 65.0 | 72.1 | 70.4 | 85.4 | 59.6 |
| 4% | **1441.9** | 321.4 | **63.4** | 64.6 | 71.3 | 69.3 | 85.3 | 59.6 |
| 8% | 1433.0 | **322.1** | 63.2 | **65.1** | 72.2 | **70.6** | **85.6** | 59.9 |
| 16% | 1434.3 | 281.8 | **63.4** | 64.8 | **72.3** | 70.1 | 84.4 | 59.5 |
| 32% | 1431.3 | 317.1 | 63.1 | 64.7 | 72.0 | 70.1 | 85.2 | 59.7 |

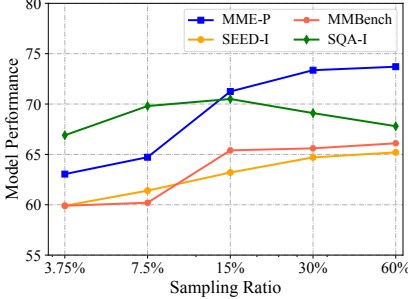

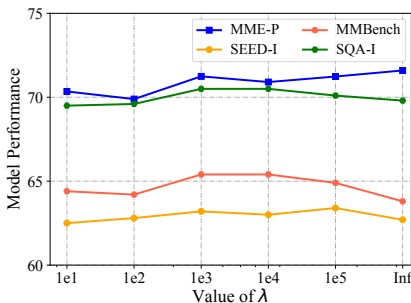

(a) The ablation of different sampling ratio.

(b) The ablation of selecting data with different hyperparameter $\lambda$.

Figure 4: The results of ablation study about the data size and hyperparameter $\lambda$.

model's performance increases at first and then stabilizes. A possible reason for this is that MME-P tends to evaluate the model's ability on visual recognition while the other two benchmarks focus on the model's general reasoning capability. Furthermore, We find that the model can maintain a certain level of performance under the minimal data size, indicating that models can acquire basic capability for downstream tasks even with a minimal amount of data.

**Influence of Different Warm-up Data Size.** We design a series of experiments to investigate the influence of different warm-up data size on the performance of TIVE. We simply change the sampling ratio for warm-up data and maintain consistency in other parts of TIVE selection. The results are presented in Table 5.

As we can observe, as the sampling ratio increases, the model performance initially exhibits a slight increase trend. Then, it begins to oscillate when the sampling ratio reaches 8%. Even so, the performance differences between various sampling ratios are quite minimal. This implies that a reference model trained with a minimal amount of warm-up data is already effective for TIVE.

**Influence of Different Hyperparameter $\lambda$.** To achieve a balanced choice between data effectiveness and data diversity, we introduce a hyperparameter $\lambda$ to control the temperature of weight distribution. We study the influence of different $\lambda$ on the quality of final selected data. We set $\lambda$ to different values and evaluate the model's performance on downstream benchmarks.

The evaluation results on MME-P, MMBench and SQA-I are shown in Figure 4b. We observe a consistent slight increase in the model's performance on MME-P benchmark as $\lambda$ increases, indicating that the MME-P benchmark is highly sensitive to instruction diversity, which is consistent with previous conclusions. On the other hand, the performance on SQA-I and MMBench initially increases with the escalation of $\lambda$, then shows a decline once the $\lambda$ reaches $1e3$. The results demonstrate that our approach with $\lambda = 1e3$ is an optimal data selection strategy that balances data effectiveness and data diversity for the model's consistent optimal performance across all downstream tasks.

## 5 RELATED WORK

**Visual Instruction Tuning.**   Visual instruction tuning is a crucial part of the construction of LVLMs, which aims to enhance the model's ability on instruction following. The collection of visual instructions is essential for visual instruction tuning. Early studies often employ LLMs to synthesize visual instructions. LVLMs trained on these instructions demonstrate promising capabilities in visual conversation and instruction following, but fail to achieve satisfactory performance on academic benchmark (Goyal et al., 2017; Schwenk et al., 2022; Marino et al., 2019). Subsequent studies (Liu et al., 2023a; Luo et al., 2024; Dai et al., 2023) have usually mixed the synthesized visual instructions and instructions from existing academic datasets together as the final instruction data. LVLMs trained on these mixtures of instructions demonstrate exceptional performance in both understanding and generation scenarios. Despite the success, these efforts solely combine all instructions in a simple way, neglecting the potential redundancy within the instructions from different tasks. We investigate the redundancy in existing visual instruction datasets and propose a measurement for data value based on instance influence and task difficulty to reduce redundancy.

**Data Selection for Instruction Tuning.**   With the advancement of LLMs, the significance of data selection has become increasingly prominent due to the high training costs. As for instruction tuning, LIMA (Zhou et al., 2024) is the first to demonstrate that instruction tuning can be accomplished with only a small amount of data. Chen et al. (2023a) further explores the potential of low data usage in task-specific models. Subsequent efforts focus on estimating the importance of an instruction sample. The importance can be estimated based on certain prior characteristics (*e.g.* length, complexity, diversity) (Liu et al., 2023c; Cao et al., 2023), with the assistance of language models (Jain et al., 2023; Liu et al., 2023c; Li et al., 2023c), by human efforts (Zhuo et al., 2024; Muennighoff et al., 2023), or using the gradient-based influence estimation on the validation set of the target benchmark (Xia et al., 2024). Compared to the data selection approach for language instruction tuning, our approach doesn't only rely on prior characteristics of texts, but considers the importance of visual instructions from a holistic perspective of both image and text. Compared to LESS (Xia et al., 2024), our approach doesn't require data from downstream benchmark, thereby achieving better generalization ability.

**Data Selection for Visual Instruction Tuning.**   Fewer studies have been focusing on data-efficient visual instruction tuning. To the best of our knowledge, there are only two studies currently conducted in this area. Among these studies, InstructionGPT-4 (Wei et al., 2023) selects high-quality instructions based on several metrics designed in their studies and SELF-FILTER (Chen et al., 2024) proposes selecting instruction data with higher diversity and difficulty by training a score-net. Compared to these studies, We are the first to study data selection for a highly complex mixture of visual task instructions, which provides much better results than the candidate datasets from these studies. To handle such complex visual instructions, we propose a gradient-based approach to estimate data value for efficient and effective task learning. With our approach, we accomplish better results compared to previous studies on data selection for visual instruction tuning with our selected data.

## 6 CONCLUSION

In this work, we focus on the redundancy issue within a mixture of visual instruction datasets that have been widely used for fine-tuning LVLMs. Through our empirical studies, we find that a significant redundancy exists in the mixed visual instruction datasets, with varying redundancy degrees across different task instructions. To eliminate redundancy, we design a novel method namely TIVE, which first estimates data value based on instance influence and task difficulty, then determines the instruction task proportion and selects representative instances to compose a smaller visual instruction subset for training. Experimental results indicate that, with the help of our data selection method, using only about 15% data can achieve comparable performance as the full-data fine-tuned model across eight benchmarks, even surpassing it on some of the benchmarks.

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

# A    OVERALL TIME COST

We present the overall time cost for TIVE and TIVE-efficient at 15% sampling ratio compared to full-data fine-tuning in Table 6:

Table 6: Statistics of base training data for empirical studies.

| Steps | TIVE | TIVE-efficient | Full-data training |
|---|---|---|---|
| Warm-up training | 0.9h | 0.9h | - |
| Gradient computation | 9.6h | 1.0h | - |
| Predicting data value | - | 0.6h | - |
| Data value estimation | 0.2h | 0.2h | - |
| Visual instruction tuning | 1.7h | 1.7h | 11.5h |
| Total | 12.4h | 4.4h | 11.5h |

For TIVE's original implementation, we can see that although the time cost for visual instruction tuning (1.7h) is much lower compared to full-data training, the total time cost is slighter higher than full-data sft. However, for TIVE-efficient, the total time cost of TIVE is significantly lower compared to full-data training. This achieves our goal of data-efficient visual instruction tuning.

# B    TRAINING DATASET

## B.1    TRAINING DATASET FOR EMPIRICAL ANALYSIS

The visual instruction dataset used for our empirical analysis is a subset of the original LLaVA-1.5 instructions. We select one dataset for each type of task. More specifically, We select VQAv2 (Goyal et al., 2017) dataset for Open-Ended Visual Question Answering (OE-VQA), A-OKVQA dataset (Schwenk et al., 2022) for Multi-Choice Visual Question Answering (MC-VQA), RefCOCO (Mao et al., 2016; Kazemzadeh et al., 2014) dataset for Referring Expression Comprehension (REC), LLaVA-1.0 (Liu et al., 2023b) dataset for Visual Conversation (VC), CC3M (Sharma et al., 2018) dataset for Image Caption (IC), and ShareGPT (Zheng et al., 2023) dataset for Textual Conversation (TC). Appendix B contains details about the datasets. The details of these selected datasets are as followed:

- *Open-Ended Visual Question Answering (OE-VQA):* it requires a model to generate natural language answers without predefined options. We select VQAv2 (Goyal et al., 2017) since it's one of the most commonly-used OE-VQA dataset.

- *Multi-Choice Visual Question Answering (MC-VQA):* it also requires a model to answer visual questions, but only selects the answer from the provided candidate choices. We select the A-OKVQA dataset (Schwenk et al., 2022).

- *Referring Expression Comprehension (REC):* it requires a model to generate the regional description of the given object or select the correct object based on the given description. We select RefCOCO dataset (Mao et al., 2016; Kazemzadeh et al., 2014).

- *Visual Conversation (VC):* it requires a model to generate long conversations based on visual content. We select the VC data from instructions of LLaVA-1.0 (Liu et al., 2023b).

- *Image Caption (IC):* it requires a model to provide an description of the given image. We select CC3M dataset (Sharma et al., 2018) as it is already used for cross-modal alignment training of LLaVA-1.5 (Liu et al., 2023a).

- *Textual Conversation (TC):* it requires the model to generate conversation in a text-only setting. We select ShareGPT (Zheng et al., 2023), as it has been widely used in training LLMs.

The statistics of our base dataset are presented in Table 7.

Table 7: Statistics of base training data for empirical studies.

| Task | MC-VQA | OE-VQA | REC | VC | Caption | TC |
|------|--------|--------|-----|-----|---------|-----|
| Numbers. | 60K | 80K | 120K | 40K | 100K | 40K |

### B.2 TRAINING DATASET FOR MAIN EXPERIMENTS

We conduct experiments on three datasets. In the experiment of evaluating TIVE against other baseline methods, we adopt the LLaVA-1.5 instruction datasets. In the experiment of evaluating the transferability of TIVE across different datasets, we additionally use the Mini-Gemini and SVIT-Mix instruction datasets. LLaVA-1.5 and SVIT-Mix contains over 600K instructions and Mini-Gemini contains over 1.4M instructions. All these datasets encompass at least nine sub-task datasets. We exclude the caption data from the selection process since it's already been trained during the LLaVA's pre-training stage.

## C EVALUATION BENCHMARKS

To comprehensively evaluate the efficacy of our approach, we evaluate TIVE across various benchmarks. The details of these benchmarks are as followed:

- *MME:* (Fu et al., 2023) it evaluates LVLM's reasoning ability from the two dimensions of perception and cognition. Each instance in MME includes an image and two binary questions. We evaluate TIVE on both splits.

- *MMBench:* (Liu et al., 2023d) it is a systematically-constructed dataset for evaluating the capacity of LVLMs. It encompasses an evaluation of 20 fine-grained capabilities of LVLMs. The evaluation is performed through its official website. We evaluate TIVE on both english split and chinese split to test its multilingual capability.

- *SEED-Bench:* (Li et al., 2023a) it develops a comprehensive set of multimodal evaluation tasks across twelve dimensions with the assistence of GPT-4. SEED-Bench encompasses assessments of both image and video understanding capabilities. In our experiments, we only utilize the image benchmark of SEED-Bench.

- *ScienceQA:* (Lu et al., 2022) it is a benchmark constructed around various science topics, encompassing both pure text-based questions and image-related text questions. In our experiment, we assess ScienceQA under both multi-modal and uni-modal setting.

- *POPE:* (Li et al., 2023d) it designs a polling-based query approach for the evaluation of object hallucination. It contains 3000 binary questions and support four evaluation metrics. In our experiment, we report the results of accuracy.

For simplicity, we only adopt MME-Perception, MMBench, and ScienceQA-Image during our empirical analysis.

## D BASELINE

We compare our methods with several baselines for data selection: (1) *Random Selection* selects data randomly; (2) *Instruction Length* utilizes length of instruction to determine the importance of an instruction sample; (3) *Perplexity* computes the perplexity score of an instruction sample to measure its importance; (4) *GraNd* (Paul et al., 2021) measures the importance of each sample by the L2-norm of the gradient caused by each sample; (5) *EL2N* (Paul et al., 2021) measures the importance of each sample by the L2-norm of the error vector of each sample. (6) *MoSo* (Tan et al., 2024) determines the importance of one sample by measuring how empirical risk changes when the sample is removed from the original training dataset. (7) *LESS* (Xia et al., 2024) determines the importance of one sample by computing the similarity of gradients from this sample and the validation samples. The EL2N scores are primarily used for estimating sample importance in image classification tasks. To adapt it for visual instruction tuning, we compute all the error vectors for

---

**Algorithm 1** Estimating Task Difficulty and Instance Influence.

---

**Require:** Instruction dataset $\mathcal{D} = \{D_1, ..., D_n\}$ ;
1: Training a reference model $M_\theta$;
2: **for** $D_i \in \mathcal{D}$ **do**
3:   Initialize task difficulty $v_i^t \leftarrow 0$ ;
4:   Initialize task gradient $g_i^t \leftarrow 0$ ;
5:   **for** $s_j \in D_i$ **do**
6:     $v_i^t \leftarrow v_i^t + \nabla l(s_j, \theta) \cdot \nabla l(s_j, \theta)$ ;                          // Self-influence.
7:     $g_i^t \leftarrow g_i^t + \nabla l(s_j, \theta) / |\nabla l(s_j, \theta)|$ ;              // Adding up instance gradient.
8:   **end for**
9:   **for** $s_j \in D_i$ **do**
10:     Final instance influence $v_s^j \leftarrow \nabla l(s_j, \theta) \cdot g_i^t / |D_i| |\nabla l(s_j, \theta)|$ ;
11:   **end for**
12:   Final task difficulty $v_i^t \leftarrow v_i^t / |D_i|$ ;
13: **end for**
14: **return** $v^t, v^i$

---

each token in each sample, and then compute the final EL2N score by averaging norms of all error vectors.

We also compare TIVE with other baseline models in Figure 1 and Table 1. These models include: BLIP-2 (Li et al., 2023b), InstructBLIP-7B (Dai et al., 2023), Shikra (Chen et al., 2023b), IDEFICS-80B (Laurençon et al., 2024), Qwen-VL (Bai et al., 2023), Qwen-VL-Chapt (Bai et al., 2023), InstructionGPT-4 (Wei et al., 2023), SELF-FILTER (Chen et al., 2024), Yi-VL-34B (Young et al., 2024), LLaVA-Next-8B (Liu et al., 2024), Libra (Xu et al., 2024), and DeepSeek-VL (Lu et al., 2024).

# E    IMPLEMENTATION DETAILS

We utilize Bunny (He et al., 2024) to construct LVLM with different LLM backbone. The LVLMs include LLaVA-1.5-7B, LLaVA-1.5-13B, LLaVA-Phi-3-4B, and LLaVA-LLaMA3-8B. The training datasets include LLaVA-1.5 instructions, SVIT-Mix (Zhao et al., 2023a) instructions and, Mini-Gemini (Li et al., 2024) instructions. Since LLaVA-1.5-7B and LLaVA-1.5-13B takes Vicuna-7B and Vicuna-13B as their LLM backbone, we denote these two models as LLaVA-Vicuna-7B and LLaVA-Vicuna-13B in some experiments. We follow the training settings of LLaVA-1.5 across all experiments. During fine-tuning, the learning rate is set to 2e-5 and the batch size is set to 16. All models are trained for one epochs. The training settings for reference models are the same as the previous settings. For all experiments, we sample 8% of the total instructions and train the reference model on the sampled data for one epoch. For TIVE-efficient, we only sample 10% of samples for gradient computation and use LLaVA-qwen1.5-1.8B for predicting influences, while for TIVE we compute the gradients for all samples. We provide a detailed description of TIVE in Algorithm 1 and Algorithm 2.

# F    ADDITIONAL EXPERIMENT DETAILS

## F.1    CHARACTERISTICS OF COMPUTED TASK PROPORTION

We present the task proportion calculated via the task-level data value for LLaVA-1.5 instructions in Table 8.

As we can observe, the VQA data (both general and multi-choice) takes up the major proportion of the selected subset, followed by grounding data, whereas conversation-related data occupies the smallest proportion. This proves that tasks related to visual perception are the most difficult in visual instruction tuning, while visual conversation is relatively simple, and text conversation is the easiest since language models are already capable of tackling such tasks. These findings indicate

---

**Algorithm 2** Data Selection Based on Data Value.

---

**Require:** Instruction dataset $\mathcal{D} = \{D_1, ..., D_n\}$ ;
**Require:** Data Value $v^t, v^i$, pruning ratio $\delta$ ;
1: Initialize target dataset $\mathcal{D}_\mathcal{T} \leftarrow \{\}$;
2: **for** $D_i \in \mathcal{D}$ **do**
3:   Determine data proportion $|D_i'| = |\mathcal{D}|v_i^t / \sum_{j=1}^n v_j^t$ ;
4:   Map weight distribution $w_i = \mathrm{softmax}(v^i / \lambda)$ ;
5:   $D_i' = \mathrm{Sample}_{w_i}(D_i, |D_i'|)$ ;                    // Sample $D_i'$ based on weights $w_i$.
6:   Merge into the target data $\mathcal{D}_\mathcal{T} \leftarrow \mathcal{D}_\mathcal{T} + D_i'$ ;
7: **end for**
8: **return** $\mathcal{D}_\mathcal{T}$

---

Table 8: Statistics of calculated task proportion.

| Task | General VQA | Multi-Choice VQA | Grounding | Visual Conversation | Text Conversation |
|---|---|---|---|---|---|
| Proportion | 20.0% | 12.7% | 12.8% | 35.3% | 4.1% |

that the central difficulty in visual instruction tuning still lies in endowing LLMs with the ability to comprehend visual content, hence, the weight of this type of data should be increased.

Interestingly, in our findings presented in section 2, we discover that increasing the amount of certain task(such as General VQA, Multi-choice VQA) data significantly improve the model's performance, while scaling other task data(such as visual conversation and text conversation) did not. The tasks for which data scaling notably enhance model performance align with the most difficult tasks estimated based on our proposed task difficulty. This consistency also validates the effectiveness of our approach.

### F.2 TIVE'S PERFORMANCE AT DIFFERENT SAMPLING RATIO

To verify whether TIVE can exhibits enhanced performance at higher sampling rates and to identify the optimal sampling rate. We conduct extensive experiments of TIVE at various sampling rates, including 15%, 30%, 50%, and 100%. The results are presented in Table 9.

As we can observe, at a sampling rate of 50%, TIVE exhibits the most superior average task performance, achieving comparable or better performance relative to the full-data baseline across nearly all benchmarks. Furthermore, we discover that the performance of the model on different downstream benchmarks varies with increasing sampling rates. We observe a consistent performance improvement on MME-P and SEED-I, while on MMBench and SQA-I, the model's performance exhibits a trend of initial increase followed by a decline. We posit that this phenomenon is attributable to the characteristics of the downstream tasks. For tasks that demand more on visual perception (such as MME-P and SEED-I ), the benefits of improved visual perception capability from increased data size outweigh the negative impact of redundancy. However, for tasks that demand more on inference (such as MMBench and SQA-I), a small amount of data can help the model learn basic inference patterns in visual scenarios while the risk of potential overfitting caused by increased data size may interfere with its inference process, causing a significant negative impact. From the perspective of average performance across all tasks, a sampling rate of around 50% appears to be ideal. However, in practical scenarios, the optimal choice of sampling rate needs to consider the specific task type, as well as the trade-off between performance and time cost.

### F.3 TRANSFERABILITY OF DATA SUBSETS SELECTED BY TIVE ACROSS DIFFERENT MODELS

In this section, we discuss whether TIVE is model-specific by evaluating if the data subsets selected by TIVE can transfer to different models. In our experiments, we select data subsets through a series of smaller models (after warm-up training) and conduct training on a larger model, LLaVA-Vicuna-13B. We present the results in Table 10.

Table 9: Evaluation results of TIVE's performance at different sampling ratio.

| Sampling rate | MME-P | MMBench | SEED-I | SQA-I | Avg. |
|---|---|---|---|---|---|
| 15% | 1433.0 | 65.0 | 63.2 | 70.6 | 67.6 |
| 30% | 1477.2 | 66.5 | 64.6 | 70.8 | 68.9 |
| 50% | 1506.1 | 66.7 | 66.2 | 69.6 | 69.3 |
| 100% (baseline) | 1510.7 | 64.3 | 66.1 | 66.8 | 68.2 |

Table 10: Evaluation results of TIVE's transferability across different models.

| Method | MME-P | MMBench | SEED-I | SQA-I | Avg. |
|---|---|---|---|---|---|
| Random | 1465.6 | 64.9 | 63.4 | 69.4 | 67.6 |
| Length | 1445.4 | 62.8 | 63.2 | 69.8 | 67.0 |
| TIVE (LLaVA-Vicuna-7B) | 1498.2 | 66.1 | 64.7 | 72.0 | 69.4 |
| TIVE (LLaVA-Phi-3-4B) | 1488.4 | 64.8 | 64.0 | 71.4 | 68.7 |
| TIVE (LLaVA-LLaMA3-8B) | 1503.3 | 65.6 | 63.6 | 71.8 | 69.0 |
| TIVE (LLaVA-Vicuna-13B) | 1502.9 | 66.1 | 65.6 | 72.2 | 69.8 |

The results indicate that these data subsets are actually transferrable. Although they perform slightly worse than the data subsets selected based on the same model (LLaVA-Vicuna-13B), they still significantly outperform other baseline methods. This suggests that TIVE is not entirely model-specific. When computational resources are extremely limited, selecting data on a smaller model and transferring it to other larger models is an efficient alternative.

### F.4    Scaling Instruction Numbers across all models

To further explore the trend of model performance as data size changes, we conduct the experiments of scaling selected instructions on more models. The data selection approach is still consistent with the previous experiments. The results are presented in Figure 5.

We find that the performance of different models follow a similar trend with the increase in instruction number. When the sampling rate is low (less than 15%), the performance of all models significantly improves with the increase in instruction number. However, when the sampling rate reaches 15%, the model's performance gradually stabilizes, scaling instruction number will have minimal effect to the

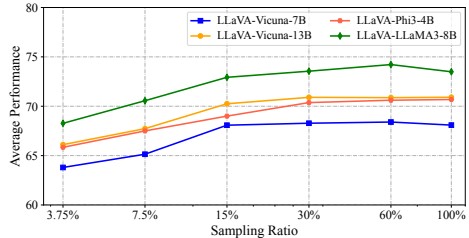

Figure 5: The experiment of scaling instruction number on different LVLMs.

model's performance. Meanwhile, when the sampling rate exceeds 60%, increasing the number of instructions can even have a negative impact on some models. These experimental results indicate that visual instruction redundancy is clearly present in different models and can potentially have a significant side effect.

## G    Limitation

### G.1    TIVE's scalability to other instruction datasets

Our experiments on three datasets demonstrate the effectiveness of TIVE on instruction datasets ranging from 0.5M to 2M. Ideally, our method can be adapted to any visual instruction dataset. However, practically, the computational cost of gradient calculations for extremely large instruction datasets (reaching the scale of 10M or even 100M) is substantial, necessitating a more efficient implementation (even more efficient than TIVE-efficient) to make the application of TIVE on these instruction datasets feasible. Simultaneously, when our target instruction dataset encompasses a vast number of tasks (over 1000), and the differences between each task are not particularly distinct, the effectiveness of our approach warrants further exploration.

## G.2 TIVE'S APPLICABILITY TO DIFFERENT TYPES OF MULTIMODAL TASKS

Although We have evaluate TIVE across multiple downstream benchmarks, we still acknowledge the insufficiency in our understanding TIVE's impact on the model's generalized performance across different tasks. Intuitively, the influence of data redundancy on the learning of various model capabilities is different. For highly visual-related abilities, such as entity recognition or OCR, the learning of these abilities necessitates a large amount of data. In this case, pruning the original instruction dataset through TIVE could easily lead to a decline in these specific abilities. Conversely, for some reasoning tasks, as these abilities primarily derive from LLMs, they do not require a significant amount of data for learning. Therefore, the removal of redundancy has little to no impact on these abilities, and may even yield improvements. Our method eliminates redundancy from a comprehensive perspective (aiming at improving average overall performance), but overlooks how to establish an optimal parameter (such as sampling rate, temperature) for dataset selection in specific scenarios, with the aim of retaining the most effective specific capabilities.

## G.3 OTHER POTENTIAL LIMITATION

Firstly, theoretically speaking, our method can be applied to any training scenario, not just limited to visual instruction tuning. However, we have not sufficiently discussed these types of scenarios in our paper. Secondly, we address the redundancy issue in a large dataset composed of a vast amount of highly diverse instruction data combinations. but our approach necessitates the use of existing task labels to categorize these different data points. For current instruction dataset, most of these labels are accessible and correct, so TIVE is able to effectively reduce redundancy on both task-level and instance-level. However, in other scenarios where these labels are either non-existent or inaccurate, we may need to manually categorize these different data using methods such as clustering. In such circumstances, whether TIVE remains effective worths further exploration. Thirdly, given the requirement for gradient computation, TIVE entails a relatively high time cost. Although we can optimize the calculation of task difficulty and instance influence with TIVE-efficient, these strategies may introduce a certain degree of estimation deviation. While these losses are relatively minor, there is still room for better implementation.

