# OpenReview forum: "LESS IS MORE: HIGH-VALUE DATA SELECTION FOR VISUAL INSTRUCTION TUNING"
_ICLR.cc/2025/Conference — Submitted to ICLR 2025_

### Official Review · Reviewer_g2yS · 2024-10-28

**Soundness:** 2
**Presentation:** 3
**Contribution:** 3
**Rating:** 5
**Confidence:** 4

**Summary:**

This paper focuses on the issue of data redundancy in visual instruction tuning for building large vision language models (LVLMs). Through empirical studies, it reveals significant redundancy within visual instruction datasets and proposes a high-value data selection approach named TIVE. TIVE estimates instance influence scores and task difficulty scores based on gradient-based influence functions to select a representative subset of data, reducing training costs while achieving comparable performance to full-data fine-tuned models on multiple benchmarks.

**Strengths:**

1. The paper addresses a crucial problem in the field of LVLMs data redundancy in visual instruction tuning. This is an important issue as it can lead to increased training costs and potential overfitting.
2. The authors conduct a series of experiments to demonstrate the existence of data redundancy and the effectiveness of their proposed method. The analysis includes pruning the amount of instructions from different tasks and evaluating the performance on various benchmarks, providing strong evidence for their claims.
3. The proposed TIVE method is innovative, considering both instance influence and task difficulty scores for data selection. This holistic approach takes into account the characteristics of different tasks and instances, making it more effective than traditional data selection methods.

**Weaknesses:**

1. Lack of in-depth analysis of task characteristics: Although the paper considers task difficulty in its method, it could provide a more in-depth analysis of the characteristics of different tasks and how they affect data redundancy and model performance.
2. Why does Instance Influence consider the gradients of other samples (Formula 1)? Is it for the normalized comparison of all samples in Formula 2?
3. A significant limitation of the current study is the lack of ablation experiments to evaluate the relative importance of instance selection versus task selection. The paper would be strengthened by comparing the proposed method against two baseline scenarios: one using only instance influence for global selection without task-level grouping, and another applying task selection first followed by random sampling or established methods like GraNd within each selected task. Such comparisons would help quantify the individual contributions of these two selection mechanisms and provide stronger justification for the proposed two-stage approach.

**Questions:**

Is there any example analysis to see how the samples that were filtered out compare to those that were retained?

---

> ### Author Response · Authors · 2024-11-20
> **Official Response to Reviewer g2yS (Part 1/2)**
>
> We are sincerely thankful for the reviewer's insightful observations and suggestions. We will address these points in the sections below.
>
> > [W1 & Q1] Although the paper considers task difficulty in its method, it could provide a more in-depth analysis of the characteristics of different tasks and how they affect data redundancy and model performance. Is there any example analysis to see how the samples that were filtered out compare to those that were retained?
>
> Following the suggestion of the reviewer, we provide the proportion of each task within the selected subset after task-level reweighting. This proportion is demonstrated in the table below.
>
> | Task       | General VQA | Multi-Choice VQA | Grounding | Visual Conversation | Text Conversation |
> | ---------- | ----------- | ---------------- | --------- | ------------------- | ----------------- |
> | Proportion | 45.5%       | 35.3%            | 11.4%     | 5.5%                | 2.2%              |
>
> As we can observe, the VQA data (both general and multi-choice) takes up the major proportion of the selected subset, followed by grounding data, whereas conversation-related data occupies the smallest proportion. This proves that tasks related to visual perception are the most difficult in visual instruction tuning, while visual conversation is relatively simple, and text conversation is the easiest since language models are already capable of tackling such tasks. These findings indicate that the central difficulty in visual instruction tuning still lies in endowing LLMs with the ability to comprehend visual content, hence, the weight of this type of data should be increased.
>
> Interestingly, in our empirical study, we discover that increasing the amount of certain task(such as General VQA, Multi-choice VQA) data significantly improve the model's performance, while scaling other task data(such as visual conversation and text conversation) did not. The tasks for which data scaling notably enhance model performance align with the most difficult tasks estimated based on our proposed task difficulty. This consistency also validates the effectiveness of our approach.
>
> > [W2-1] Why does Instance Influence consider the gradients of other samples? (Formula 1)
>
> Formula 1 offers a standard algorithm[1, 2] for computing the influence of an instance $s$ on another instance $s'$. This influence formulation measures how training on one instance $s$ impacts the model's loss on another instance $s'$, which is accomplished by calculating the dot product of the two instances' gradients. When the gradients of the two instances are on a similar direction, training on one instance would correspondingly assist the learning of the other instance. Therefore, we need to compute the gradients of other samples for influence estimation. Despite this, the instance influence we propose is not the influence between instances as described in formula 1, but rather **the influence of an instance on its corresponding task**. Therefore, we need to compute the influence of the target instance on all other samples within the same task, average them, and then obtain the instance-task influence, as shown in Formula 2. The result value is used as the instance influence for data selection.
>
> **Reference:**
>
> [1] Pruthi, Garima, et al. "Estimating training data influence by tracing gradient descent." *Advances in Neural Information Processing Systems* 33 (2020): 19920-19930.
>
> [2] Park, Sung Min, et al. "Trak: Attributing model behavior at scale." *arXiv preprint arXiv:2303.14186* (2023).

---

> ### Author Response · Authors · 2024-11-20
> **Official Response to Reviewer g2yS (Part 2/2)**
>
> > [W2-2] Is it for the normalized comparison of all samples in Formula 2?
>
>
> $$
> v^i_{s} = \frac{1}{|D_i|}\sum_{s' \in D_i \setminus s} \frac{\nabla l(s, \theta) \cdot \nabla l(s', \theta)}{|\nabla l(s, \theta)| |\nabla l(s', \theta)|}.
> $$
>
>
> We present formula 2 above. For the numerator $\nabla l(s, \theta) \cdot \nabla l(s', \theta)$ in the summation function, we compute the influence of instance $s$ on all other instances. We do this to estimate the influence of $s$ on its corresponding task, as described in our response of the previous question. As for the denominator $|\nabla l(s, \theta)| |\nabla l(s', \theta)|$, it is for the purpose of normalization.  We do it to mitigate the potential impact of instances with excessively large gradients on the final computation of influence, thereby avoiding introducing large amount of noisy data.
>
>
>
> > [W3] A significant limitation of the current study is the lack of ablation experiments to evaluate the relative importance of instance selection versus task selection. The paper would be strengthened by comparing the proposed method against two baseline scenarios: one using only instance influence for global selection without task-level grouping, and another applying task selection first followed by random sampling or established methods like GraNd within each selected task. Such comparisons would help quantify the individual contributions of these two selection mechanisms and provide stronger justification for the proposed two-stage approach.
>
> | Benchmarks | Both | Only Task-level | Only Instance-level w/ ETG | Only Instance-level w/o ETG | Neither |
> | ---------- | ---- | --------------- | -------------------------- | --------------------------- | ------- |
> | SQA-I      | 70.6 | 69.8            | 68.2                       | 67.5                        | 68.4    |
> | MMB        | 65.0 | 63.7            | 62.9                       | 62.6                        | 62.5    |
> | SEED-I     | 63.2 | 62.7            | 62.9                       | 62.3                        | 62.2    |
>
> Thanks for your valuable feedback! We sincerely apologize for the unclear presentation, but we have indeed conducted the experiments in table 4 and present related analysis in the ablation studies. In the experiments, we evaluate the efficacy of task-level value by select data based on instance influence, while set the proportion of all tasks in the target data subset to a fixed value(all tasks are consider equal regardless of their difficulty). The efficacy of instance-level value is verifies by computing task weights based on task difficulty, but select instances within the task data randomly. In response to the reviewer's suggestions , we have supplemented our experiments by focusing solely on data selection at the instance level without considering task proportions. We present the results in the table above. In the table, ETG indicates "equal task grouping", meaning if we set a fixed proportion for equal task grouping, or just select data completely on instance-level. The above results indicate that both task-based and instance-based selection contribute to improvement in the final model's performance. However, data selection based on task difficulty only contributes more substantially to the reduction of redundancy compared to selection based on instance influence only.  This suggests that in the current visual instruction data, redundancy between tasks is more pronounced than between instances, hence reducing redundancy at the task level proves to be more effective.
>
> We would like to thank the reviewer again for your insightful comments and constructive criticism. Your feedback has greatly contributed to improving the quality of our work. We hope that our responses have adequately addressed your concerns. We kindly request you to consider raising the score, and we welcome any further feedback or concerns that you might have. We are more than willing to engage in further discussions to clarify any remaining issues.

---

> ### Author Response · Authors · 2024-11-25
>
> Dear Reviewer g2yS,
>
> Thank you for the time and effort you generously invest reviewing our manuscript. We've tried to carefully address your concerns in our response. We hope that our detailed response, the supplementary experiments, and the revised version of our manuscript can successfully address your concern.
>
> As the discussion phase is drawing to a close, we would be appreciative if you could spare some time to go over our response. If our responses have successfully addressed your concerns, would you might consider reevaluating your initial assessment and possibly adjusting the score? If any unresolved issues still exist, we are fully prepared to tackle them.
>
> Best regards,
>
> The Authors

---

> ### Author Response · Authors · 2024-11-28
>
> Dear Reviewer g2yS,
>
> Thanks for your valuable time and hard work in reviewing our paper. We've made our best effort to respond to your comments. As we're nearing the end of the discussion stage, we are writing to kindly remind you that the rebuttal period is coming to a close. We're looking forward to continuing the conversation about our responses and any other parts of our research with you. Please let us know if you have any concerns.
>
> Best regards,
>
> The Authors

---

> > ### Author Response · Authors · 2024-12-01
> >
> > Dear Reviewer g2yS,
> >
> > Thank you for your time and effort in reviewing our paper. We have addressed your comments and want to kindly remind you that the rebuttal period is ending soon. We look forward to discussing our responses and any other aspects of our research with you. Please let us know if you have any concerns.
> >
> > Best regards,
> >
> > The Authors

---

> ### Author Response · Authors · 2024-12-03
>
> Dear Reviewer g2yS,
>
> Thank you for your time and effort in reviewing our paper. We have addressed your comments and want to kindly remind you that the rebuttal period is ending soon. We look forward to discussing our responses and any other aspects of our research with you. Please let us know if you have any concerns.
>
> Best regards,
>
> The Authors

---

### Official Review · Reviewer_i5DE · 2024-10-31

**Soundness:** 2
**Presentation:** 2
**Contribution:** 2
**Rating:** 5
**Confidence:** 3

**Summary:**

The paper investigates redundancy within visual instruction datasets used for fine-tuning large vision-language models (LVLMs). It presents an empirical analysis which reveals that reducing the instruction data does not significantly impact model performance, suggesting the potential for data reduction. To address this, the authors propose TIVE, a novel method that selects high-value data based on task difficulty and instance influence using gradient-based techniques. Experiments demonstrate that TIVE can achieve comparable or even superior results to full-data models while using only 15% of the dataset. The proposed method provides a more efficient approach to visual instruction tuning by minimizing training costs and redundancy.

**Strengths:**

1.  The paper introduces a well-justified and innovative method, TIVE, that addresses data redundancy in visual instruction datasets for LVLMs.
2. The motivation for addressing redundancy is well explained, and the proposed solution is logically developed based on detailed empirical findings.
3. The authors provide thorough empirical evidence demonstrating the existence of redundancy within current visual instruction datasets, supporting the motivation for their approach.

**Weaknesses:**

1. The paper does not sufficiently discuss the potential limitations of the TIVE approach, such as its scalability to even larger datasets or its applicability to different types of multimodal tasks.
2. I have some concerns regarding the data selection approach. In the earlier stages of machine learning, data and feature selection were widely popular. However, recent trends show that using larger models with bigger datasets tends to yield remarkable generalization capabilities. I hope the authors can address this concern in their rebuttal.

**Questions:**

Please refer to weakness part

---

> ### Author Response · Authors · 2024-11-20
> **Official Response to Reviewer i5DE (Part 1/2)**
>
> We sincerely thank the reviewer for their time and the valuable feedback provided. We will endeavor to address all the raised concerns in the subsequent sections.
>
> > [W1] The paper does not sufficiently discuss the potential limitations of the TIVE approach, such as its scalability to even larger datasets or its applicability to different types of multimodal tasks.
>
> We appreciate your insightful suggestions regarding the limitations of our methodology. We'll discuss the limitation as followed:
>
> **TIVE's scalability to even larger datasets**
>
> Our experiments on three datasets have demonstrated the effectiveness of our approach on datasets ranging from 0.5M to 2M. Ideally, our method can be adapted to any visual instruction dataset. However, practically, the computational cost of gradient calculations for larger instruction datasets (reaching the scale of 10M or even 100M) is substantial, necessitating a more efficient implementation to make the application of TIVE on these instruction datasets feasible. Simultaneously, when our target instruction dataset encompasses a vast number of tasks (over 1000), and the differences between each task are not particularly distinct, the effectiveness of our approach warrants further exploration.
>
> **TIVE's applicability to different types of multimodal tasks**
>
> In the era of LLMs (Large Language Models), it is critical for models to exhibit strong generalization abilities, meaning that after training on a specific instruction dataset, they should be able to generalize to a wide variety of distinct tasks. This is also true to LVLMs(Large Vision-Language Models). Hence, our method is independent of the downstream tasks. Moreover, the applicability of TIVE that we discuss here does not conform to the traditional setting where training and testing data fall under the same domain. In our experiments, we ensure that there is no overlap between the instruction data used for training and the downstream tasks. Despite this, we acknowledge the insufficiency in our understanding TIVE's impact on the model's generalized performance across different tasks. Intuitively, the influence of data redundancy on the learning of various model capabilities is different. For highly visual-related abilities, such as entity recognition or OCR, the learning of these abilities necessitates a large amount of data. In this case, pruning the original instruction dataset through TIVE could easily lead to a decline in these specific abilities. Conversely, for some reasoning tasks, as these abilities primarily derive from LLMs, they do not require a significant amount of data for learning. Therefore, the removal of redundancy has little to no impact on these abilities, and may even yield improvements. Our method eliminates redundancy from a comprehensive perspective, but overlooks how to establish an optimal parameter (such as sampling rate, temperature)  for dataset selection in specific scenarios, with the aim of retaining the most effective specific capabilities.
>
> **Other potential limitation**
>
> Firstly, theoretically speaking, our method can be applied to any training scenario, not just limited to visual instruction tuning. However, we have not sufficiently discussed these types of scenarios in our paper. Secondly, we address the redundancy issue in a large dataset composed of a vast amount of highly diverse instruction data combinations. but our approach necessitates the use of existing task labels to categorize these different data points. When these labels are either non-existent or inaccurate, we may need to manually categorize these different data points using methods such as clustering. In such circumstances, whether TIVE remains effective worths further exploration. Thirdly, given the requirement for gradient computation, TIVE entails a relatively high time cost. Although we can optimize the calculation of task difficulty and instance influence with existing techiniques,  these strategies may introduce a certain degree of estimation deviation. While these losses are relatively minor, there is still room for better implementation.

---

> ### Author Response · Authors · 2024-11-20
> **Official Response to Reviewer i5DE (Part 2/2)**
>
> > [W2] I have some concerns regarding the data selection approach. In the earlier stages of machine learning, data and feature selection were widely popular. However, recent trends show that using larger models with bigger datasets tends to yield remarkable generalization capabilities. I hope the authors can address this concern in their rebuttal.
>
> This is a interesting question. Existing large models, particularly large language models (LLMs), mostly have two training stages. In the first stage, the model is pre-trained on a vast amount of unsupervised data. In the second stage, the model is fine-tuned on supervised data. It is true that during the pre-training stage, using larger models with bigger datasets tends to yield remarkable generalization capabilities. This phenomenon is often referred to as the scaling law[1], where the model's generalization performance is continually improved by increasing the quantity of model parameters and training data. However, our research primarily focuses on the model's second training stage, namely supervised fine-tuning (with a specific focus on instruction tuning in our study). This stage is primarily aimed at aligning LLMs with human intent, that is, learning to follow human instruction and produce a helpful response. In this instruction tuning stage, we will not introduce new knowledge or capabilities other than human intention alignment into the model. Therefore, the scaling law is not universally applicable.
>
> Research on instruction tuning of LLMs has found that redundancy exists in language instruction data [2, 3, 4, 5]. A small amount of high-quality data can enable the model to achieve excellent alignment performance, while introducing excessive amount of data might potentially lead to overfitting[6], even impair the LLM's original generalization capabilities. For visual instruction tuning, the language model needs to learn not only human instruction following, but also visual understanding. To achieve these two targets, existing visual instruction data often combines traditional visual data (captions, VQA, grounding) with synthesized visual instruction following data [7, 8, 9, 10], which leads to increased redundancy due to this straightforward combination method.
>
> In our study, we firstly demonstrate that redundancy in visual instruction data is indeed significant through an empirical study. Subsequently, we propose a method TIVE for estimating data value based on task difficulty and instance influence. This method particularly considers the potential causes of redundancy that might exist in visual instruction data derived from multiple different sources, and performs data selection based on this estimated data value. Ultimately, our approach is proven to be effective across different visual instruction sets and various models. Therefore, we assert that it is necessary to eliminate redundancy and select high-quality data in the context of visual instruction tuning. The method we propose, which effectively eliminates redundancy in visual instruction data, also makes a significant contribution.
>
> We thank the reviewer again for your valuable feedback. We hope that our responses have sufficiently addressed your concerns. We kindly request that you consider revising the score based on these clarifications. We welcome any additional feedback or concerns you may have and are more than willing to engage in further discussions to clarify any remaining issues.
>
> **Reference:**
>
> [1] Kaplan, Jared, et al. "Scaling laws for neural language models." *arXiv preprint arXiv:2001.08361* (2020).
>
> [2] Zhou, Chunting, et al. "Lima: Less is more for alignment." *Advances in Neural Information Processing Systems* 36 (2024).
>
> [3] Liu, Wei, et al. "What makes good data for alignment? a comprehensive study of automatic data selection in instruction tuning." *arXiv preprint arXiv:2312.15685* (2023).
>
> [4] Xia, Mengzhou, et al. "Less: Selecting influential data for targeted instruction tuning." *arXiv preprint arXiv:2402.04333* (2024).
>
> [5] Li, Ming, et al. "From quantity to quality: Boosting llm performance with self-guided data selection for instruction tuning." *arXiv preprint arXiv:2308.12032* (2023).
>
> [6] Shi, Zhengyan, et al. "Instruction Tuning With Loss Over Instructions." *arXiv preprint arXiv:2405.14394* (2024).
>
> [7] Liu, Haotian, et al. "Visual instruction tuning." *Advances in neural information processing systems* 36 (2024).
>
> [8] Liu, Haotian, et al. "Improved baselines with visual instruction tuning." *Proceedings of the IEEE/CVF Conference on Computer Vision and Pattern Recognition*. 2024.
>
> [9] Zhu, Deyao, et al. "Minigpt-4: Enhancing vision-language understanding with advanced large language models." *arXiv preprint arXiv:2304.10592* (2023).
>
> [10] Zhao, Bo, et al. "Svit: Scaling up visual instruction tuning." *arXiv preprint arXiv:2307.04087* (2023).

---

> ### Author Response · Authors · 2024-11-25
>
> Dear Reviewer i5DE,
>
> Thank you for the time and effort you generously invest reviewing our manuscript. We've tried to carefully address your concerns in our response. We hope that our detailed response, the supplementary experiments, and the revised version of our manuscript can successfully address your concern.
>
> As the discussion phase is drawing to a close, we would be appreciative if you could spare some time to go over our response. If our responses have successfully addressed your concerns, would you might consider reevaluating your initial assessment and possibly adjusting the score? If any unresolved issues still exist, we are fully prepared to tackle them.
>
> Best regards,
>
> The Authors

---

> ### Comment · Reviewer_i5DE · 2024-11-26
> **Official Response**
>
> After reviewing the response and considering the comments from all other reviewers, I have decided to maintain my original negative score

---

> > ### Author Response · Authors · 2024-11-28
> >
> > We express our gratitude once again for your time and feedback. We are keen to understand any additional concerns that might have prevented you from giving a higher score. We are more than willing to further address these concerns for your satisfaction. On the other hand, as other reviewers have increased their scores, we hope you might reconsider increasing the score. We look forward to your response.

---

> > > ### Author Response · Authors · 2024-12-01
> > >
> > > Dear Reviewer i5DE,
> > >
> > > Thank you for your time and effort in reviewing our paper. We have addressed your comments and want to kindly remind you that the rebuttal period is ending soon. We look forward to discussing our responses and any other aspects of our research with you. Please let us know if you have any concerns.
> > >
> > > Best regards,
> > >
> > > The Authors

---

> ### Author Response · Authors · 2024-12-03
>
> Dear Reviewer i5DE,
>
> Thank you for your time and effort in reviewing our paper. We have addressed your comments and want to kindly remind you that the rebuttal period is ending soon. We look forward to discussing our responses and any other aspects of our research with you. Please let us know if you have any concerns.
>
> Best regards,
>
> The Authors

---

### Official Review · Reviewer_P1cf · 2024-11-04

**Soundness:** 3
**Presentation:** 3
**Contribution:** 3
**Rating:** 6
**Confidence:** 5

**Summary:**

This work studied the redundancy problem in the visual instruction tuning dataset for LVLMs. It proposed a high-value data selection approach TIVE, to eliminate redundancy within the visual instruction data and reduce the training cost. HIVE can effectively reduce the training data size of different VLM instruction tuning datasets across different models without compromising the overall performance.

**Strengths:**

The observation of the dataset redundancy problem aligns with the community's observations.
The proposed TIVE method sounds reasonable.
The authors conduct extensive experiments with detailed analysis to demonstrate the effectiveness of the method and its components.

**Weaknesses:**

1. Though the experiments are comprehensive, there are several points to further discuss or clarify in the method. See questions.
2. The authors need to provide a further discussion on the overall cost of the method: as TIVE needs the reference model trained with warmup data, the selection of TIVE is generally model-specific. TIVE needs to compute the LoRA gradient over all samples in the pool, then this cost is close to training on all of the data with LoRA. Tuning the hyper-parameters of HIVE would give another dimension of complexity if there are no default hyper-parameters. From this perspective, this method may fail to reduce the overall training costs. If so, it needs to target improving the final performance (without insisting on 15% of data) and discuss more about how to achieve this (what proportion of data is the best?). If not, the corresponding additional cost should be discussed.

**Questions:**

1. In algorithm 1, the task influence is calculated in a nested for loop, with a overall complexity $O(|D_i|^2)$ for each task. A question is, could the author first use one pass to aggregate the average of the normalized gradients and then use another pass to calculate the score? This will reduce the complexity to linear. Will this cause numerical instability or it doesn't? Originally, was the gradients stored or re-computed?
2. Are the influence scores' gradient of a sample computed over all tokens in it and do average, or only on outputs part?
3. In the line 301, $\lambda$ is introduced as "We use a hyperparameter $\lambda$ to control the temperature of the weight distribution". However, how actually it is used is presented in Line 871 in appendix. The ablation of $\lambda$ appears before readers know how actually it is used. The ordering of this part needs further consideration.

---

> ### Author Response · Authors · 2024-11-20
> **Official Response to Reviewer P1cf (Part 1/2)**
>
> We deeply appreciate the reviewer's positive review and the insightful comments. We will clarify the raised concerns in the subsequent sections.
>
> > [Q1] In algorithm 1, the task influence is calculated in a nested for loop, with a overall complexity for each task. A question is, could the author first use one pass to aggregate the average of the normalized gradients and then use another pass to calculate the score? This will reduce the complexity to linear. Will this cause numerical instability or it doesn't? Originally, was the gradients stored or re-computed?
>
> Thanks for your feedback on approach details. The answer to the first question is yes. In fact, the influence computation based on a double-loop is equivalent to calculating the average gradient through one pass first, then computing the influence, and this does not result in numerical instability. As for the third question, all gradients are pre-computed and stored through one pass. We use LoRA training and random projection to reduce the dimensionality of these gradients, thus the subsequent storage cost of the gradient features and the computation cost of gradient-based influence are both much lower compared to the computation of gradients.
>
>
> ---
> > [Q2] Are the influence scores' gradient of a sample computed over all tokens in it and do average, or only on outputs part?
>
> This is a interesting question. We follow the mainstream visual instruction tuning approaches [1, 2, 3] and only compute gradients on the output tokens.
>
> ---
>
> > [Q3] In the line 301, $\lambda$ is introduced as "We use a hyperparameter to control the temperature of the weight distribution". However, how actually it is used is presented in Line 871 in appendix. The ablation of  $\lambda$  appears before readers know how actually it is used. The ordering of this part needs further consideration.
>
> We greatly appreciate your suggestions regarding the presentation in our manuscript. We sincerely apologize for not providing a proper explanation of the term $\lambda$ in the methodology section. We will ensure to rectify this in the revised version.
>
>
> ---
> > [W3-1] As TIVE needs the reference model trained with warmup data, the selection of TIVE is generally model-specific.
>
> | Method                    | MME-P  | MMBench | SEED-I | SQA-I | Avg. |
> | ------------------------- | ------ | ------- | ------ | ----- | ---- |
> | Random                    | 1456.6 | 64.9    | 63.4   | 69.4  | 67.6 |
> | Length                    | 1445.4 | 62.8    | 63.2   | 69.8  | 67.0 |
> | TIVE from LLaVA-Vicuna-7B | 1498.2 | 66.1    | 64.7   | 72.0  | 69.4 |
> | TIVE from LLaVA-Phi-3-4B  | 1488.4 | 64.8    | 64.0   | 71.4  | 68.7 |
> | TIVE from LLaVA-LLaMA3-8B | 1503.3 | 65.6    | 63.6   | 71.8  | 69.0 |
> | TIVE                      | 1502.9 | 66.1    | 65.6   | 72.2  | 69.8 |
>
> In our experiments, we indeed need to train a unique reference model for each individual model. The reference model should have the same backbone as the target model, so the selection of TIVE is model-specific. However, upon further analysis, we discover that the selected data subset is actually transferrable.  For instance, the data selected based on the LLaVA-Vicuna-7B reference model is actually applicable to LLaVA-Vicuna-13B or LLaVA-LLaMA3-8B as well. We present the results in the table above, where we select data subsets through a series of smaller models and conduct training on a larger model, LLaVA-Vicuna-13B. The results indicate that these data subsets are actually transferrable. Although they perform slightly worse than the data subsets selected based on the same model(LLaVA-Vicuna-13B), they still significantly outperform other baseline methods. This suggests that TIVE is not entirely model-specific. When computational resources are extremely limited, selecting data on a smaller model and transferring it to other larger models is an efficient alternative.
>
>
> **Reference:**
>
> [1] Liu, Haotian, et al. "Visual instruction tuning." *Advances in neural information processing systems* 36 (2024).
>
> [2] Liu, Haotian, et al. "Improved baselines with visual instruction tuning." *Proceedings of the IEEE/CVF Conference on Computer Vision and Pattern Recognition*. 2024.
>
> [3] Zhu, Deyao, et al. "Minigpt-4: Enhancing vision-language understanding with advanced large language models." *arXiv preprint arXiv:2304.10592* (2023).

---

> ### Author Response · Authors · 2024-11-20
> **Official Response to Reviewer P1cf (Part 2/2)**
>
> > [W3-2] TIVE needs to compute the LoRA gradient over all samples in the pool, then this cost is close to training on all of the data with LoRA. From this perspective, this method may fail to reduce the overall training costs.
>
> Thanks for your insightful suggestion. In our original TIVE implementation, we do have to compute the LoRA gradients over all samples and the overall cost of TIVE is slightly higher than full-data fine-tuning. To address this, we propose an efficient implementation of TIVE, TIVE-efficient, where we only randomly select 10% of samples and compute their gradients. We obtain the task difficulty based on these gradients and compute their instance influence. After that, we train a smaller model, LLaVA-Qwen-1.5-1.8B, for predicting instance influence. In this way, we have slightly compromise the estimation precision for task difficulty and instance influence, but significantly reduce TIVE's time cost. we present the time cost and evaluation results of TIVE-efficient in the below table. As we can observe, TIVE-efficient substantially reduce the time cost of TIVE, without any significant compromise in performance.
>
> |                | Total Time Cost |
> | -------------- | --------------- |
> | TIVE           | ~ 12.6h         |
> | TIVE-efficient | ~ 4.6h          |
> | Full-data SFT  | ~ 11.5h         |
>
> | Method             | MME-P      | MMBench  | SEED-I   | SQA-I    | Avg.     |
> | ------------------ | ---------- | -------- | -------- | -------- | -------- |
> | Random             | 1386.5     | 61.8     | 61.9     | 68.4     | 65.4     |
> | Length             | 1413.0     | 59.3     | 61.2     | 69.2     | 65.1     |
> | Perplexity         | 1393.3     | 62.3     | 61.3     | 67.9     | 65.3     |
> | GraNd              | 1400.5     | 62.9     | 62.3     | 68.4     | 65.9     |
> | EL2N               | 1356.5     | 61.6     | 61.9     | 66.2     | 64.5     |
> | **TIVE**           | **1433.0** | **65.0** | **63.2** | **70.6** | **67.6** |
> | **TIVE-efficient** | **1424.9** | **64.3** | **62.5** | **70.8** | **67.2** |
>
>
> ---
> > [W3-3] Tuning the hyper-parameters of TIVE would give another dimension of complexity if there are no default hyper-parameters.
>
> Yes, finding the optimal hyper-parameters can introduce additional time cost. Therefore, we present the impact of these hyperparameters on model performance in our ablation study. We hope that these results can assist the selection of optimal hyperparameters, thereby reducing the associated time cost.
>
>
> ---
> > [W3-4] From this perspective, this method may fail to reduce the overall training costs. If so, it needs to target improving the final performance (without insisting on 15% of data) and discuss more about how to achieve this (what proportion of data is the best?). If not, the corresponding additional cost should be discussed.
>
> | Sampling rate  | MME-P  | MMBench | SEED-I | SQA-I | Avg. |
> | -------------- | ------ | ------- | ------ | ----- | ---- |
> | 15%            | 1433.0 | 65.0    | 63.2   | 70.6  | 67.6 |
> | 30%            | 1477.2 | 66.5    | 64.6   | 70.8  | 68.9 |
> | 50%            | 1506.1 | 66.7    | 66.2   | 69.6  | 69.3 |
> | 100%(baseline) | 1510.7 | 64.3    | 66.1   | 66.8  | 68.2 |
>
> Firstly, we have discussed the additional cost and the way to reduct cost, see our response above. Secondly, without insisting on 15% sampling rate, our models **can be further enhanced**. In the above table, we present the results of TIVE at various sampling rates, including 15%, 30%, 50%, and 100%. At a sampling rate of 50%, TIVE exhibits the most superior average task performance, achieving comparable or better performance relative to the full-data baseline across nearly all benchmarks. Furthermore, we discover that the performance of the model on different downstream benchmarks varies with increasing sampling rates. We observe a consistent performance improvement on MME-P and SEED-I, while on MMBench and SQA-I, the model's performance exhibits a trend of initial increase followed by a decline. We posit that this phenomenon is attributable to the characteristics of the downstream tasks. For tasks that demand more on visual perception (such as MME-P and SEED-I ), the benefits of improved visual perception capability from increased data size outweigh the negative impact of redundancy. However, for tasks that demand more on inference (such as MMBench and SQA-I), a small amount of data can help the model learn basic inference patterns in visual scenarios while the risk of potential overfitting caused by increased data size may interfere with its inference process, causing a significant negative impact. From the perspective of average performance across all tasks, a sampling rate of around 50% appears to be ideal. However, in practical scenarios, the optimal choice of sampling rate needs to consider the specific task type, as well as the trade-off between performance and time cost.

---

> ### Author Response · Authors · 2024-11-20
>
> We thank again for the time and effort that the reviewer has invested in evaluating our study. We hope that our response have resolved the raised concerns. We respectfully ask for a reconsideration of the score in light of these responses.  Please let us know if you have any further feedback or concerns. We are more than willing to engage in further discussions to clarify any remaining issues.

---

> > ### Comment · Reviewer_P1cf · 2024-11-21
> >
> > Thanks to the authors for the detailed response, which clarifies my points of concern. I would like to encourage the authors to include these discussions and results in the paper to make this work more solid and complete (especially how Tive-efficient can save cost, how original Tive can be used to improve performance; the linear implementation to avoid unnecessary complexity; also other points to clarify).
> >
> > As there are still several days, I would encourage the authors to include the mentioned revision (and mark the changes with color). I may raise my score according to the adjusted content. Overall, I hold a positive view of this paper。

---

> > > ### Author Response · Authors · 2024-11-22
> > >
> > > Thank you for your positive feedback and for providing us with valuable suggestions to further improve our work. We have carefully considered your feedbacks and incorporated the revisions in the updated manuscript. We hope that these revisions can address your concerns and strengthen the paper. We are very grateful for your constructive suggestions and your positive view of our work.
> > >
> > > If you have any further feedback or additional concerns, we are happy to address them. Thank you again for your time and consideration.

---

> > > > ### Comment · Reviewer_P1cf · 2024-11-22
> > > > **One thing to remind about the resivison**
> > > >
> > > > One thing to remind: you should not use the final print format in the revision and show the authors' names.

---

> > > > > ### Author Response · Authors · 2024-11-22
> > > > >
> > > > > Thank you for your reminder. We have updated the revision to the correct format.

---

> ### Author Response · Authors · 2024-11-25
>
> Dear Reviewer P1cf,
>
> Thank you for the time and effort you generously invest reviewing our manuscript. We've tried to carefully address your concerns in our response. We hope that our detailed response, the supplementary experiments, and the revised version of our manuscript can successfully address your concern.
>
> As the discussion phase is drawing to a close, we would be appreciative if you could spare some time to go over our response. If our responses have successfully addressed your concerns, would you might consider reevaluating your initial assessment and possibly adjusting the score? If any unresolved issues still exist, we are fully prepared to tackle them.
>
> Best regards,
>
> The Authors

---

> > ### Comment · Reviewer_P1cf · 2024-11-25
> >
> > Thanks to the authors for the careful revision and follow-ups. The initial 6 points are to encourage meaningful improvement; if the cost-performance trade-off is not solved, this work will fall below the threshold in the final review. Currently, the quality of this work has improved substantially and is above the acceptance threshold, and I hold a positive view. I will increase my confidence score to 5 and suggest considering acceptance, while I may not raise the score further, according to the work quality distribution of ICLR.
> >
> > (For follow-up work, if the best amount of data for instruction finetuning could be further studied, not only empirically but also insightfully why it is that ratio, I would rate that kind of work with 8 or even 10. But that is a harder problem.)

---

> > > ### Author Response · Authors · 2024-11-28
> > >
> > > Thank you immensely for your active participation and valuable insights during the discussion phase. Your feedback is crucial and has greatly aided in improving the quality of our paper. We will continue to refine our research based on your suggestions, even if we cannot further revise the paper. However, we have a point of confusion. As the reviewer points out, finding the optimal amount of data for visual instruction tuning is very important. Yet, this heavily depends on the actual computation budget and the specific target task, making it difficult to identify an optimal sampling rate to achieve the desired effect. Can we interpret this as, for a specific target task, is it possible to find an optimal sampling rate that saturates the model's performance?

---

> > > > ### Comment · Reviewer_P1cf · 2024-11-28
> > > >
> > > > Partially yes.
> > > > The more interesting thing here is how we can better understand the data construction. For the unselected samples, are these samples non-beneficial at the beginning? Or if otherwise, why given some other samples presented, these samples are no longer beneficial? Are there methods to measure this influence and mitigate this problem, so that we can always construct a better dataset?
> > > > These are the harder open questions; and for the data selection problem, are there some theoretical, even heuristics methods that can help us predict the performance saturation?
> > > > If the authors have good thoughts on these questions, I am more than happy to have further discussion.

---

> > > > > ### Author Response · Authors · 2024-11-29
> > > > >
> > > > > Thank you for your suggestions. We are delighted to share some of our insights：
> > > > >
> > > > > 1. For the unselected samples, are these samples non-beneficial at the beginning?
> > > > >
> > > > > Our understanding is that the majority of the unselected data is also effective. We define effectiveness as whether a sample contributes positively to the overall task learning, which is measured by the instance-task influence. In our experiments, the vast majority of samples indeed have a positive influence, indicating that they are effective. However, due to sampling rate limitations, we can only select the most effective portion of the samples.
> > > > >
> > > > > 2. Or if otherwise, why given some other samples presented, these samples are no longer beneficial?
> > > > >
> > > > > This is a interesting question. In fact, given the other samples presented (as our selected samples) and trained, these unselected samples indeed become less beneficial as training progress. Our main finding is, most samples have very similar effect (measured via instance influence) on model training, even if they look like completely different samples. This results in a situation where if sample A and sample B initially contribute equally to task learning, the contribution of sample B to task learning will decrease after training on sample A, as the model has already learned the knowledge contained in this sample. This leads to the phenomenon that if we further train on B, the model's learning efficacy on this task would be minimal, and could even result in negative effects of overfitting on such samples. That's why we need to select a effective portion of samples to reduce redundancy.
> > > > >
> > > > > 3. Are there methods to measure this influence and mitigate this problem, so that we can always construct a better dataset?
> > > > >
> > > > > In our opinion, the redundancy issues exist in all datasets, with similar data points present in almost every dataset. These data points can always be learned from a small subset of samples, and training on all data would result in redundancy. In the era of large models, this problem becomes increasingly apparent. Given that large models inherently encompass extensive world knowledge and generalization capabilities, many data points may not provide any gain to the model and could be considered redundant. However, for more challenging tasks, the issue of redundancy tends to be alleviated. Our proposed TIVE strives to address the redundancy issue from this perspective. Our response to question 2 actually explain why we need to select the most effective samples, and why training on these subset might have better performance on the entire task dataset. However, as we notice, if a task is really hard, then the unselected samples might even still be beneficial after training on the selected subset.  If we still consider the issue of sample A and sample B which contribute similarly to task learning. Suppose the overall task is very challenging, then even after training on sample A, the model still struggles to fully learn from such samples. At this point, sample B's contribution to task learning indeed decrease, but not to near zero (compared to less challenging tasks), indicating that it is still an effective sample. Therefore, we need to consider increasing the proportion of samples for this task, sampling more of these effective samples to learn such difficult tasks. This is also the reason we introduce the task-difficulty based value in our study.
> > > > >
> > > > > 4. Are there some theoretical, even heuristics methods that can help us predict the performance saturation?
> > > > >
> > > > > This is a really hard problem. From the perspective of training dynamics, indeed, we can predict whether the model's training has saturated, i.e., whether the loss has ceased to decrease. This can be estimated through influence estimation. If the influence of sample A on sample B is zero or negative, then training on sample A will not effectively reduce the model's loss on sample B. As our paper primarily focuses on improving the model's generalization, we did not use data from downstream tasks. However, if we aim to precisely estimate the model's performance on a specific downstream task, we would need to calculate the influence of the remaining samples on the downstream task data at every stage of the model's training. If no sample point effectively reduces the loss on the downstream task, then the model's training has already saturated. Although this approach is theoretically feasible, in practice, it requires repeated gradient calculations, leading to significantly higher computational costs. Moreover, a reduction in loss does not necessarily result in performance gains for the actual task and may even introduce potential overfitting.

---

> > > > > > ### Comment · Reviewer_P1cf · 2024-12-03
> > > > > >
> > > > > > Glad to see your thoughts on these questions. For the next version of your paper, I would suggest including some of these insights in the discussion. And wish you good luck!

---

### Official Review · Reviewer_WdBE · 2024-11-04

**Soundness:** 2
**Presentation:** 3
**Contribution:** 3
**Rating:** 6
**Confidence:** 4

**Summary:**

The paper concentrates on reducing the data redundancy of instruction-following MLLMs. The authors show that pruning a certain ratio of specific training data has a slight influence on the overall accuracy. Based on the observation, the authors present a data selection approach named TIVE. The data selection strategy is based on estimating task difficulty and instance influence. Then the gradient features are used for selection. The authors integrate the approach on several MLLM backbones including LLaVA-1.5, LLaVA-LLaMA3, Mini-Gemini, etc. Experiments on multimodal benchmarks show an increase compared with random selection.

**Strengths:**

1. The experiments are complete across various MLLM backbones, including Vicuna, Phi, and LLaMA3, and architectures, including LLaVA-1.5, SVIT-Mix, and Mini-Gemini. The authors also show comparisons with baselines / advanced MLLMs.
2. The performance meets the full baselines with only 10% to 30% training data, which shows the effectiveness of TIVE.
3. The paper is well written and the formulation periods are clear.

**Weaknesses:**

1. The main weakness lies in the design of the approach, especially regarding the computation costs. In my recognition, the inference operation based on the gradients and other selection operations are costly, even meets the original training cost. This makes the contribution of the pruning method weak.
2. The selection based on gradients is a posterior probability, which means choosing the hard samples as prior knowledge. This may be unfair for the comparisons against baselines.
3. The overall performance in Table 1 against backbone models is weak, only shows significant improvement on the SciQA benchmark, and gains accuracy drop or fair on other benchmarks (may be due to experiment uncertainty). This may mean the selection approach is sub-optimal.

**Questions:**

See the weakness part. The authors are encouraged to answer such questions.
1. Regarding weakness 1, the authors are encouraged to provide the actual time cost for TIVE and fair comparisons with full training for LLaVA-1.5.
2. The accuracy drop for TIVE is significant compared with the baseline.
3. The authors could explain the design of gradient inference in detail and show the relations with the original training data and the backbone.
Therefore, I recommend rejecting this manuscript in its current version. I would like to increase my score if the authors could address the issues above or provide more results against approaches with similar targets.

---

> ### Author Response · Authors · 2024-11-20
> **Official Response to Reviewer WdBE (Part 1/4)**
>
> We sincerely thank the reviewer for their comprehensive review and helpful feedback. We will try to address your concerns below.
>
> > [W1 & Q1] The main weakness lies in the design of the approach, especially regarding the computation costs. In my recognition, the inference operation based on the gradients and other selection operations are costly, even meets the original training cost. This makes the contribution of the pruning method weak. Regarding this weakness, the authors are encouraged to provide the actual time cost for TIVE and fair comparisons with full training for LLaVA-1.5.
>
> We appreciate your insightful suggestions regarding our methodology and understand your concern about the total time cost. We provide the actual time cost of TIVE and the comparsion of  TIVE's overall time cost to full fine-tuning in the table below (on 8 x A100 80G GPUs).
>
> |                     | Total Time Cost                     |
> | ------------------- | ----------------------------------- |
> | TIVE                | ~ 0.9h + 9.6h + 0.2h + 1.7h = 12.4h |
> | Full-data fine-tune | ~ 11.5h                             |
>
> As can be observed, warm-up training,  gradient computation, task difficulty and instance influence estimation, and visual instruction tuning on selected subset spends approximately **0.9h**, **9.6h**, **0.2h**, **and** **1.7h** respectively. Although the time cost for visual instruction tuning(1.7h) is much lower compared to full-data training, the total time cost is slightly higher than full-data sft.
>
> Despite this, our work also has potential contributions to the field. Firstly, we verify the existence of redundancy in visual instruction data and effectively reduce this redundancy using a new proposed approach.  Secondly, there are many existing methods that can be employed to accelerate TIVE [1, 2, 3, 4, 5]. For example, we can select a small subset of samples, compute their gradients, and use them to estimate task difficulty, instead of using the whole dataset. Besides, we can also use these computed influences to train a smaller model for predicting the instance influences of the remaining samples, instead of computing influences based on the large model.
>
> We sincerely thank the reviewer for mentioning the efficiency issue of TIVE. To address it, **we propose a more efficient implementation of TIVE, TIVE-efficient** here. Firstly, We **only select 10% of samples and compute their gradients**. Then, we obtain the task difficulty based on these gradients and compute their instance influence. After that, we train a small model, LLaVA-Qwen-1.5-1.8B, for predicting instance influence of other samples. In this way, we slightly compromise the estimation precision for task difficulty and instance influence, but greatly reduce TIVE's time cost. we present the total time cost and evaluation results of TIVE-efficient in the table below.
>
> |                     | Total Time Cost                           |
> | ------------------- | ----------------------------------------- |
> | TIVE                | ~ 0.9h + 9.6h + 0.2h + 1.7h = 12.4h       |
> | TIVE-efficient      | ~ 0.9h + 1.0h + 0.6h + 0.2h + 1.7h = 4.4h |
> | Full-data fine-tune | ~ 11.5h                                   |
>
>
> **Reference:**
>
> [1] Kobayashi, Sosuke, et al. "Efficient estimation of influence of a training instance." *arXiv preprint arXiv:2012.04207* (2020).
>
> [2] Kwon, Yongchan, et al. "DataInf: Efficiently Estimating Data Influence in LoRA-tuned LLMs and Diffusion Models." *The Twelfth International Conference on Learning Representations*.
>
> [3] Guu, Kelvin, et al. "Simfluence: Modeling the influence of individual training examples by simulating training runs." *arXiv preprint arXiv:2303.08114* (2023).
>
> [4] Influence tuning: Demoting spurious correlations via instance attribution and instance-driven updates
>
> [5] Zhou, Kun, et al. "JiuZhang3. 0: Efficiently Improving Mathematical Reasoning by Training Small Data Synthesis Models." *arXiv preprint arXiv:2405.14365* (2024).

---

> ### Author Response · Authors · 2024-11-20
> **Official Response to Reviewer WdBE (Part 2/4)**
>
> We provide a continuation of our previous response to [W1 & Q1] here. We present the results of TIVE and TIVE-efficient below.
> | Method             | MME-P      | MMBench  | SEED-I   | SQA-I    | Avg.     |
> | ------------------ | ---------- | -------- | -------- | -------- | -------- |
> | Random             | 1386.5     | 61.8     | 61.9     | 68.4     | 65.4     |
> | Length             | 1413.0     | 59.3     | 61.2     | 69.2     | 65.1     |
> | Perplexity         | 1393.3     | 62.3     | 61.3     | 67.9     | 65.3     |
> | GraNd              | 1400.5     | 62.9     | 62.3     | 68.4     | 65.9     |
> | EL2N               | 1356.5     | 61.6     | 61.9     | 66.2     | 64.5     |
> | **TIVE**           | **1433.0** | **65.0** | **63.2** | **70.6** | **67.6** |
> | **TIVE-efficient** | **1424.9** | **64.3** | **62.5** | **70.8** | **67.2** |
>
> As we can observe, the performance of TIVE-efficient is similar to the original TIVE, and consistently outperforms other models. These experiments confirm that TIVE can be implemented in a more efficient way, without significant compromise in performance. We will supplement the design details and experimental results of TIVE-efficient in the revised version of our paper. We hope that this efficient implemetation can address your concern on the total cost of TIVE.
>
> ---
>
> > [W2] The selection based on gradients is a posterior probability, which means choosing the hard samples as prior knowledge. This may be unfair for the comparisons against baselines.
>
> | Method         | MME-P      | MMBench  | SEED-I   | SQA-I    | Avg.     |
> | -------------- | ---------- | -------- | -------- | -------- | -------- |
> | Random         | 1386.5     | 61.8     | 61.9     | 68.4     | 65.4     |
> | Length         | 1413.0     | 59.3     | 61.2     | 69.2     | 65.1     |
> | Perplexity     | 1393.3     | 62.3     | 61.3     | 67.9     | 65.3     |
> | GraNd          | 1400.5     | 62.9     | 62.3     | 68.4     | 65.9     |
> | EL2N           | 1356.5     | 61.6     | 61.9     | 66.2     | 64.5     |
> | **MoSo**       | **1410.2** | **62.6** | **62.4** | **68.1** | **65.9** |
> | **LESS**       | **1415.1** | **63.0** | **62.2** | **68.8** | **66.1** |
> | TIVE           | 1433.0     | 65.0     | 63.2     | 70.6     | 67.6     |
> | TIVE-efficient | 1424.9     | 64.3     | 62.5     | 70.8     | 67.2     |
>
>
> We appreciate the reviewer's constructive suggestions regarding our experimental results. In fact, in the baselines we adopt, E2LN, GraNd, and Perplexity also utilize prior knowledge, and TIVE still achieves better performance compared to them.
> To make our results more convincing, we include two additional gradient-based data selection methods, MoSo[6] and LESS[7], as baselines. MoSo measures how empirical risk changes when a specific sample is removed from the original training dataset based on computed gradients, and LESS matches the gradient of training samples with validation sample to find the most effective data points. The supplementary experimental results are displayed in the table above. The results indicate that even compared to methods that also employ gradient features, TIVE and TIVE-efficient consistently achieve superior performance across almost all benchmarks.  Actually, our advantage mainly lies in our consideration on data redundancy from both task-level and instance-level, while previous studies typically focus solely on the instance-level, overlooking the variations in task difficulty.
>
> **Reference:**
>
> [1] Kobayashi, Sosuke, et al. "Efficient estimation of influence of a training instance." *arXiv preprint arXiv:2012.04207* (2020).
>
> [2] Kwon, Yongchan, et al. "DataInf: Efficiently Estimating Data Influence in LoRA-tuned LLMs and Diffusion Models." *The Twelfth International Conference on Learning Representations*.
>
> [3] Guu, Kelvin, et al. "Simfluence: Modeling the influence of individual training examples by simulating training runs." *arXiv preprint arXiv:2303.08114* (2023).
>
> [4] Influence tuning: Demoting spurious correlations via instance attribution and instance-driven updates
>
> [5] Zhou, Kun, et al. "JiuZhang3. 0: Efficiently Improving Mathematical Reasoning by Training Small Data Synthesis Models." *arXiv preprint arXiv:2405.14365* (2024).
>
> [6] Tan, Haoru, et al. "Data pruning via moving-one-sample-out." *Advances in Neural Information Processing Systems* 36 (2024).
>
> [7] Xia, Mengzhou, et al. "Less: Selecting influential data for targeted instruction tuning." *arXiv preprint arXiv:2402.04333* (2024).

---

> ### Author Response · Authors · 2024-11-20
> **Official Response to Reviewer WdBE (Part 3/4)**
>
> > [W3 & Q2] The overall performance in Table 1 against backbone models is weak, only shows significant improvement on the SciQA benchmark, and gains accuracy drop or fair on other benchmarks (may be due to experiment uncertainty). This may mean the selection approach is sub-optimal.
>
> * Performance of TIVE and TIVE-efficient compared to other baselines at 50% sample rate
> | Method             | MME-P      | MMBench  | SEED-I   | SQA-I    | Avg.     |
> | ------------------ | ---------- | -------- | -------- | -------- | -------- |
> | LLaVA-1.5          | 1510.7     | 64.3     | 66.1     | 66.8     | 68.2     |
> | Random             | 1458.2     | 63.2     | 63.8     | 68.2     | 67.0     |
> | GraNd              | 1462.6     | 63.8     | 63.2     | 67.6     | 66.9     |
> | LESS               | 1488.4     | 64.6     | 64.0     | 69.2     | 68.0     |
> | **TIVE**           | **1506.1** | **66.7** | **66.2** | **69.6** | **69.4** |
> | **TIVE-efficient** | **1500.8** | **66.3** | **66.1** | **69.8** | **69.3** |
>
> * Total time cost for TIVE and TIVE-efficient at 50% sample rate
>
> |                     | Total Time Cost                           |
> | ------------------- | ----------------------------------------- |
> | TIVE                | ~ 0.9h + 9.6h + 0.2h + 5.7h = 16.6h       |
> | TIVE-efficient      | ~ 0.9h + 1.0h + 0.6h + 0.2h + 5.7h = 8.4h |
> | Full-data fine-tune | ~ 11.5h                                   |
>
> To highlight the data redundancy issue in the dataset, we set a rather aggressive data pruning rate (85%) in our paper. In traditional machine learning settings, pruning about 50% of the original dataset would almost certainly result in a significant decrease in model performance even on simple tasks such as image classfication [1, 2, 3, 4, 5]. In contrast, we can also use lower rate to guarantee the performance. To validate it and address the reviewer's concern, we conduct experiments at a lower pruning rate (50%). We present the results and the overall cost of TIVE in this setting in the tables above. Under this setting, both TIVE and TIVE-efficient achieve similar or better results on almost all benchmarks compared to full-data fine-tune, and outperforms other data selection baselines by a large margin. In addition to this, at the pruning rate of 50%, TIVE-efficient still has a lower overall time cost compared to full-data fine-tune, but achieves better results. Interestingly, as the pruning rate decreases (from 85% to 50%), the performance gap between TIVE-efficient and TIVE becomes smaller. This might indicate that TIVE-efficient can be a better alternative of TIVE with we are allowed to retain a relatively large amount of samples.
>
> **Reference:**
>
> [1] Toneva, Mariya, et al. "An empirical study of example forgetting during deep neural network learning." *arXiv preprint arXiv:1812.05159* (2018).
>
> [2] Coleman, Cody, et al. "Selection via Proxy: Efficient Data Selection for Deep Learning." *International Conference on Learning Representations*. 2020
>
> [3] Paul, Mansheej, Surya Ganguli, and Gintare Karolina Dziugaite. "Deep learning on a data diet: Finding important examples early in training." *Advances in neural information processing systems* 34 (2021): 20596-20607.
>
> [4] Killamsetty K, Sivasubramanian D, Ramakrishnan G, et al. Glister: Generalization based data subset selection for efficient and robust learning[C]//Proceedings of the AAAI Conference on Artificial Intelligence. 2021, 35(9): 8110-8118.
>
> [5] Baldock R, Maennel H, Neyshabur B. Deep learning through the lens of example difficulty[J]. Advances in Neural Information Processing Systems, 2021, 34: 10876-10889.

---

> ### Author Response · Authors · 2024-11-20
> **Official Response to Reviewer WdBE (Part 4/4)**
>
> > [Q3] The authors could explain the design of gradient inference in detail and show the relations with the original training data and the backbone.
>
> Yes. TIVE takes as a Language-Visual Language Model (LVLM) and a visual instruction dataset $\mathcal{D}$ as input, with the aim of selecting a high-quality subset $\mathcal{D_T}$ from the original dataset  $\mathcal{D}$ without compromising performance. Firstly, we sample a small fraction of data from the original instruction data (utilizing the entire dataset is feasible, but it incurs additional computational cost). Then, this subset of data is used to train a reference model via LoRA[1], which we refer to as warm-up training. LoRA introduces trainable light-weight modules within each layer of the model while freezing the parameters of other components. We incorporate LoRA into the LLM counterpart of the LVLM, as it accounts for the largest proportion of parameters. The purpose of warm-up training is to enable the reference model to be warm-up to learn the visual instruction following capability, and will not be overfitted to the distribution of the whole visual instruction dataset. After training the reference model,  we compute gradients for the original visual instruction dataset. Specifically, we compute the gradient for each sample on each LoRA module, then concatenate the gradients from all LoRA modules. After that, we reduce the dimensionality of gradient features via random projection. We can easily pre-compute and store the gradient features since they have low dimension. Subsequently, based on these precomputed gradients, we calculate task difficulty and instance influence, and use these two estimates for data selection.
>
> We would like to express our sincere gratitude once again for the insightful comments and suggestions provided by the reviewers. We hope that our responses have adequately addressed your concerns. We kindly request you to consider raising the score, and we welcome any further feedback or concerns that you might have. We are more than willing to engage in further discussions to clarify any remaining issues.
>
> **Reference:**
>
> [1] Hu, Edward J., et al. "Lora: Low-rank adaptation of large language models." *arXiv preprint arXiv:2106.09685* (2021).

---

> ### Author Response · Authors · 2024-11-25
>
> Dear Reviewer WdBE,
>
> Thank you for the time and effort you generously invest reviewing our manuscript. We've tried to carefully address your concerns in our response. We hope that our detailed response, the supplementary experiments, and the revised version of our manuscript can successfully address your concern.
>
> As the discussion phase is drawing to a close, we would be appreciative if you could spare some time to go over our response. If our responses have successfully addressed your concerns, would you might consider reevaluating your initial assessment and possibly adjusting the score? If any unresolved issues still exist, we are fully prepared to tackle them.
>
> Best regards,
>
> The Authors

---

> ### Comment · Reviewer_WdBE · 2024-11-27
> **Official Comment by Reviewer WdBE**
>
> I appreciate the authors' efforts in improving the manuscript, particularly in the design of efficiency.  They have addressed most of my concerns, so I will raise my score.  However, I cannot give a higher score because the authors have not demonstrated significant improvements in motivations and methods compared to previous approaches in data selection.

---

> > ### Author Response · Authors · 2024-11-28
> >
> > Thank you for your suggestions and positive recognition of our work. We will continue to seek the most optimal data selection method and further optimize our paper.

---

### Author Response · Authors · 2024-11-22
**General Response**

We would like to express our gratitude once again to the reviewers for their insightful feedback and suggestions. We have uploaded a revised version of our paper, in which the modifications are highlighted in blue for your convenience.. The main modifications are:

1. In Section 3, we add the design details of TIVE-efficent (WdBE, P1cf). We also provide a more precise definition of the hyperparameter $\lambda$ in advance (P1cf) and the efficient linear implementation for computing instance influence (P1cf).
2. In Section 4, we provide the experimental results of TIVE-efficient (WdBE, P1cf) and the additional baseline results of MoSo and LESS in the main table (WdBE).
3. In Section 4, we re-present our ablation study on relative importance of instance-level selection versus task-level selection. Additionally, we add the results of TIVE using only Instance influence without any task grouping (g2yS).
4. In appendix A, we provide the detail time cost of TIVE and TIVE-efficient compared to full-data training (WdBE, P1cf).
5. In appendix E, we update the two-pass linear algorithm for computing instance-task influence in Algorithm 1 (P1cf).
6. In appendix F, We first present the task proportion of data subset selected by TIVE and conduct a thorough analysis (g2yS). Subsequently, we demonstrate the detail results of how the model's performance change on various downstream tasks at a higher sampling rate (WdBE, P1cf). Finally, We incorporate experiments evaluating the transferability of the data subsets selected by TIVE across different models.
7. In appendix G, we incorporate a discussion on the limitations of our approach (i5DE).

We sincerely thank the reviewers for your effort in reviewing this paper and hope this revised version can better address your concerns.

---

### Meta-Review · Area_Chair_WCGw · 2024-12-05

**Metareview:**

### **Summary**
This paper introduces **TIVE**, a method designed to reduce redundancy in visual instruction datasets by leveraging **task difficulty** and **instance influence** scores calculated via gradient-based techniques. The approach is evaluated on multiple vision-language models (LVLMs) and benchmarks, claiming comparable or superior performance to full-data fine-tuning with only 15% of the dataset.

### **Strengths**
1. **Problem Relevance**:
   - The issue of data redundancy in visual instruction datasets is a timely and important topic in scaling LVLMs.
2. **Empirical Validation**:
   - The authors provide extensive experimental results across various LVLM architectures and benchmarks.
3. **Efficiency Improvements**:
   - The introduction of **TIVE-efficient** shows an attempt to reduce computational costs, which is a positive step toward scalability.

### **Weaknesses**
1. **Insufficient Novelty and Impact**:
   - The approach, while novel in combining task- and instance-level selection, does not offer significant improvements over baseline methods.
   - In some benchmarks, performance gains are marginal or nonexistent, raising concerns about the practical utility of such aggressive data pruning.

2. **Scalability and Practicality**:
   - The computational overhead of TIVE remains high, especially for larger datasets, even with TIVE-efficient. The applicability to datasets larger than those tested (e.g., >10M samples) is unclear and largely speculative.
   - The reliance on gradient-based influence functions makes the approach resource-intensive, potentially negating the claimed efficiency benefits in realistic scenarios.

3. **Limited Task Insights**:
   - Although the method incorporates task difficulty, the paper does not provide a deep understanding of how task characteristics influence redundancy or the success of TIVE.
   - The empirical analysis lacks qualitative insights into the differences between selected and filtered data, leaving questions about the robustness of the selection criteria.

4. **Methodological Concerns**:
   - The proposed method relies heavily on prior knowledge of task labels and task-specific gradients, which may not generalize to more diverse or unlabeled datasets.
   - Comparisons to baselines are not entirely convincing, as the method’s advantage appears to depend on specific configurations and datasets.

5. **Presentation Issues**:
   - Key concepts (e.g., gradient-based influence, task difficulty) are not clearly explained, making the methodology difficult to follow.
   - The ablation studies, while improved during the rebuttal, do not provide enough evidence to justify the two-stage approach over simpler alternatives.

### **Recommendation**
The proposed solution lacks sufficient novelty, scalability, and practical impact to justify acceptance. The marginal improvements and methodological limitations do not demonstrate a compelling advancement over existing methods. Addressing these issues in future work, particularly by improving scalability and providing deeper insights into task-specific effects, would strengthen the paper’s contribution.

**Additional Comments On Reviewer Discussion:**

The reviewers raised several concerns, many of which were not fully resolved during the rebuttal period:
- **Scalability**: While TIVE-efficient attempts to address computational concerns, the method’s cost-effectiveness compared to full-data fine-tuning remains questionable.
- **Task Insights**: The authors acknowledged gaps in understanding task-specific effects but did not provide sufficient new insights or experiments to address this weakness.
- **Performance**: Marginal improvements on some benchmarks fail to justify the complexity and computational demands of the method.
- **Generalizability**: The approach is highly dependent on task labels and specific dataset characteristics, limiting its broader applicability.

One reviewer noted that data selection methods like TIVE may not align with current trends in leveraging larger models and datasets for better generalization. While the authors defended their approach as applicable to the instruction tuning phase, this distinction was not clearly demonstrated or validated.

---

### Decision · Program_Chairs · 2025-01-22

Reject